# *Lactiplantibacillus**plantarum* ATG-K2 Exerts an Anti-Obesity Effect in High-Fat Diet-Induced Obese Mice by Modulating the Gut Microbiome

**DOI:** 10.3390/ijms222312665

**Published:** 2021-11-23

**Authors:** Young-Sil Lee, Eun-Jung Park, Gun-Seok Park, Seung-Hyun Ko, Juyi Park, You-Kyung Lee, Jong-Yeon Kim, Daeyoung Lee, Jihee Kang, Hae-Jeung Lee

**Affiliations:** 1AtoGen Co., Ltd., Daejeon 34015, Korea; gspark@atogen.co.kr (G.-S.P.); ksh1229@atogen.co.kr (S.-H.K.); juyipark@atogen.co.kr (J.P.); luk@atogen.co.kr (Y.-K.L.); d.lee@atogen.co.kr (D.L.); jhkang@atogen.co.kr (J.K.); 2Department of Food and Nutrition, College of BioNano Technology, Gachon University, Gyeonggi-do 13120, Korea; ejpark@gachon.ac.kr (E.-J.P.); whddus95@gachon.ac.kr (J.-Y.K.); 3Institute for Aging and Clinical Nutrition Research, Gachon University, Gyeonggi-do 13120, Korea

**Keywords:** obesity, probiotics, *Lactiplantibacillus plantarum* ATG-K2, gut microbiome, short-chain fatty acids, lipid metabolism

## Abstract

Obesity is a major health problem. Compelling evidence supports the beneficial effects of probiotics on obesity. However, the anti-obesity effect of probiotics remains unknown. In this study, we investigated the anti-obesity effects and potential mechanisms of *Lactiplantibacillus plantarum* ATG-K2 using 3T3-L1 adipocytes and high-fat diet (HFD)-induced obese mice. 3T3-L1 cells were incubated to determine the effect of lipid accumulation with lysate of *L. plantarum* ATG-K2. Mice were fed a normal fat diet or HFD with *L. plantarum* ATG-K2 and Orlistat for 8 weeks. *L. plantarum* ATG-K2 inhibited lipid accumulation in 3T3-L1 adipocytes, and reduced body weight gain, WAT weight, and adipocyte size in HFD-induced obese mice, concurrently with the downregulation of PPARγ, SREBP1c, and FAS and upregulation of PPARα, CTP1, UCP1, Prdm16, and ND5. Moreover, *L. plantarum* ATG-K2 decreased TG, T-CHO, leptin, and TNF-α levels in the serum, with corresponding gene expression levels in the intestine. *L. plantarum* ATG-K2 modulated the gut microbiome by increasing the abundance of the *Lactobacillaceae* family, which increased SCFA levels and branched SCFAs in the feces. *L. plantarum* ATG-K2 exhibited an anti-obesity effect and anti-hyperlipidemic effect in 3T3-L1 adipocytes and HFD-induced obese mice by alleviating the inflammatory response and regulating lipid metabolism, which may be influenced by modulation of the gut microbiome and its metabolites. Therefore, *L. plantarum* ATG-K2 can be a preventive and therapeutic agent for obesity.

## 1. Introduction

Obesity is an extensive public health problem, and its prevalence has been increasing over the last few decades [1]. It has been known as a risk factor for chronic metabolic syndromes, such as hyperlipidemia, type 2 diabetes, and cardiovascular disease; therefore, obesity management is important to prevent obesity related complications [1]. A dietary intervention including energy restriction, macronutrients, food, and dietary intake patterns, is recommended for the treatment of obesity. Despite their effectiveness, with a dietary intervention there remains unachieved weight loss [2]. Currently, various pharmacological approaches are used to treat obesity, and their use is limited because of their undesirable side effects [3]. For these reasons, a new approach is needed to prevent and treat obesity with high safety and effectiveness, and bioactive substances, such as polyphenols, antioxidants, prebiotics (dietary fibers), and probiotics are being investigated to prevent and treat obesity.

Probiotics are live microorganisms that, when administrated in an adequate amount, confer a health benefit on the host [4], and probiotic species, Lactobacillus and Bifidobacterium, are generally regarded as safe. Several studies have reported that probiotics, including *Lactobacillus* spp., reduce obesity, insulin resistance, fatty liver, and inflammation, which exhibit the beneficial effects of probiotics on obesity and its complications [5,6,7,8]. Furthermore, the gut microbiome has recently been considered an essential regulator of host metabolism and dysbiosis or imbalance of the gut microbiome is associated with obesity and its complications [9]. Dysbiosis or an imbalanced gut microbiome means disruption of the balance between the gut microbiota and host and reflects negative shifts in abundance, diversity, and relative distribution of the gut microbiome composition [6,10]. Both in animal and human studies with obesity, an altered gut microbiome with a reduction in gut microbiome diversity and richness has been reported. Consistent findings have shown that obesity is associated with a decreased abundance in some taxa such as *Bifidobaterium*, *Christensenellaceae*, and *Akkermansia*; however, the changes of the gut microbiome that have been reported to differ with obesity have varied across studies. For example, the *Firmicutes/Bacteroidetes* ratio in an obese human compared with a lean human has been reported to decrease, to increase, or not change at all [11,12,13]. These inconsistencies may be affected by a combination of large interpersonal variation, insufficient sample sizes, and methodological differences between studies. In obese mice, a change of function, including the regulation of the production of enzymes involved in carbohydrate and lipid metabolism, has also observed. Overall, most studies have demonstrated a structural and functional dysbiosis of the gut microbiome in obesity, although there is still much debate on the role of the gut microbiome in obese humans [14,15]. Dysbiosis increases intestinal permeability and the translocation of LPS into target tissues, which causes metabolic endotoxemia. Consequently, metabolic endotoxemia induces low-grade chronic inflammation in obesity and its complications through a Toll-like receptor-mediated inflammatory pathway [16]. On the other hand, the gut microbiome plays an important role in the fermentation of carbohydrates, and they produce short-chain fatty acids (SCFAs), such as acetic, propionic, and butyric acids, which have critical roles in preventing and treating obesity and its complications by improving glucose and lipid homeostasis and decreasing inflammation [17]. Some probiotics exert metabolically beneficial effects by modulating the gut microbiome and its metabolites [14,18,19]. For example, *L. rhamnosus* LS-8 and *L. crustorum* MN407 showed an anti-obesity effect with reduced body weight gain, insulin resistance, and inflammation, but also manipulated the gut microbiota by decreasing the abundance of *Bacteroides* and *Desulfovibrio* and increasing *Lactobacillus* and *Bifdobacterium*, which led to increasing levels of SCFAs in feces [7]. Therefore, targeting changes in the gut microbiome using probiotics has emerged as a strategy.

We have previously proven *Lactiplantibacillus plantarum* ATG-K2 isolated from fermented cabbage, a traditional Korean food [20], possesses a beneficial effect on non-alcoholic fatty liver disease and intestinal inflammation in high-fat and high-fructose diet fed rats [21,22]. However, no previous study has examined the anti-obesity effects of this probiotic strain. Moreover, although there are reports that *Lactobacillus* may have potentially beneficial anti-obesity effects, the application of specific *Lactobacillus* strains for preventing and treating obesity remain limited because probiotics have species- and strain-specific effects, and the anti-obesity mechanisms have not been fully elucidated. Accordingly, in our study, we investigated the anti-obesity effect of *L. plantarum* ATG-K2 and the underlying mechanisms in vitro and in vivo.

## 2. Results

### 2.1. Effects of L. plantarum ATG-K2 on the Cell Viability of 3T3-L1 Preadipocytes and on Lipid Accumulation during 3T3-L1 Differentiation

To investigate the cytotoxic effect of *L. plantarum* ATG-K2 on 3T3-L1 cells, different *L. plantarum* ATG-K2 concentrations were exposed to 3T3-L1 cells for 72 h. As shown in Figure 1A, cell viability is not significantly affected by *L. plantarum* ATG-K2 concentrations of 25, 50, 100, 200, and 400 μg/mL. Therefore, *L. plantarum* ATG-K2 did not exhibit any cytotoxicity at the concentrations tested in this study. Next, to demonstrate the effect of *L. plantarum* ATG-K2 on lipid accumulation during 3T3-L1 differentiation, various *L. plantarum* ATG-K2 concentrations were applied when differentiation was induced, and lipid accumulation was determined by Oil Red O staining. Lipid accumulation in 3T3-L1 adipocytes is significantly reduced in a dose-dependent manner in the presence of *L. plantarum* ATG-K2 (Figure 1B,C). These results suggest that *L. plantarum* ATG-K2 suppresses lipid accumulation during 3T3-L1 differentiation.

### 2.2. Effect of L. plantarum ATG-K2 on mRNA and Protein Expression Levels Related to the Adipogenesis and Lipogenesis of 3T3-L1 Adipocytes

To investigate the effect of *L. plantarum* ATG-K2 on adipogenesis and lipogenesis, the expression of related genes was measured by RT-qPCR. Expression levels of the adipogenesis genes, PPAR*γ*, C/EBP*α*, and C/EBP*β*, and lipogenesis genes, SREBP1c, FAS, and ACC, in 3T3-L1 adipocytes were significantly decreased by treatment with the lysate of *L. plantarum* ATG-K2 compared with those in control cells (Figure 1D,E). Moreover, the relative protein levels of PPARγ, C/EBPα, SREBP1c, and FAS in *L. plantarum* ATG-K2 lysate-treated cells decreased in a dose-dependent manner (Figure 1F). These results indicate that the suppression of lipid accumulation by *L. plantarum* ATG-K2 might be associated with reduced expression levels of adipogenesis and lipogenesis genes.

### 2.3. Effect of L. plantarum ATG-K2 on the AMPK Activation of 3T3-L1 Adipocytes

To further investigate the effects of *L. plantarum* ATG-K2 on AMPK activation, we performed a Western blot analysis to detect phosphorylated AMPK and AMPK protein expression levels. AMPK activation primarily occurs during the early differentiation phase, days 1 to 3 [23]; therefore, we examined AMPK activation on day 2 in addition to day 7, which is the completion stage of differentiation. As shown in Figure 1G, AMPK activation is markedly increased during the differentiation of *L. plantarum* ATG-K2 lysate-treated cells.

### 2.4. Effects of L. plantarum ATG-K2 on Body Weight and Food Intake Rate

The body weight and body weight gain are significantly higher in the high-fat diet (HFD) group than in the normal fat diet (NFD) group after 10 weeks (Table 1). In contrast, treatment with 10 × 10^9^ CFU/day of *L. plantarum* ATG-K2 and Orlistat significantly suppresses HFD-induced body weight and body weight gain; however, these effects are not observed in the 4 × 10^9^ CFU/day of *L. plantarum* ATG-K2 treatment. Food intake in the HFD group was lower than that in the NFD and Orlistat groups. Energy intake in the NFD group was decreased compared with the HFD group, but was increased in the Orlistat group. However, the *L. plantarum* ATG-K2 group did not affect food intake and energy intake. The food efficiency ratio in the HFD group was higher than that in the NFD group; however, it was decreased by treatment with 10 × 10^9^ CFU/day of L. plantarum ATG-K2 and Orlistat compared with the HFD group, and did not decrease with 4 × 10^9^ CFU/day of *L. plantarum* ATG-K2.

### 2.5. Effects of L. plantarum ATG-K2 on White Adipose Tissue (WAT) Weight

The epididymal (Epi) WAT, perirenal (Peri) WAT, mesenteric (Mes) WAT, and total WAT weight in the HFD group are higher than those in the NFD group. They are reduced by the 10 × 10^9^ CFU/day of *L. plantarum* ATG-K2 and Orlistat treatment (Figure 2A). From the hematoxylin and eosin (H&E) analysis of Epi WAT, the size of adipocytes was considerably larger in the HFD group than in the NFD group; however, smaller adipocytes were observed and shifted in distribution towards reduced cell fat content and diameters in the 10 × 10^9^ CFU/day of *L. plantarum* ATG-K2 and Orlistat groups (Figure 2B–D). However, a reduction in WAT weight and adipocyte size was not observed at 4 × 10^9^ CFU/day of *L. plantarum* ATG-K2. The reduction of WAT weight and adipocyte size in the high dose of *L. plantarum* ATG-K2 and Orlistat groups was in parallel with a significant reduction in body weight gain by the *L. plantarum* ATG-K2 and Orlistat treatment. These results suggest that *L. plantarum* ATG-K2 decreases adiposity in HFD-induced obese mice.

### 2.6. Effects of L. plantarum ATG-K2 on Biochemical Serum Parameters, Adipokines, and Fecal Triglyceride (TG)

Treatment with 10 × 10^9^ CFU/day of *L. plantarum* ATG-K2 resulted in lower serum TG levels than in the HFD group (Table 2). Serum total cholesterol (T-CHO) levels increased by HFD were reduced by the 10 × 10^9^ CFU/day of *L. plantarum* ATG-K2 and Orlistat treatment but were not changed by the 4 × 10^9^ CFU/day of *L. plantarum* ATG-K2 treatment. Serum high-density lipoprotein cholesterol (HDL-CHO) and glucose levels were higher in the HFD group than in the NFD group but were not changed in the *L. plantarum* ATG-K2 and Orlistat groups. Serum low-density lipoprotein cholesterol (LDL-CHO) levels did not change in any of the groups. Serum adiponectin levels were decreased, while serum leptin levels were increased in the HFD group compared to the NFD group. Treatment with 10 × 10^9^ CFU/day of *L. plantarum* ATG-K2 and Orlistat significantly decreased serum leptin levels but did not affect adiponectin levels. Fecal TG showed no change in the *L. plantarum* ATG-K2 groups and significantly increased in the Orlistat group.

### 2.7. Effects of L. plantarum ATG-K2 on Lipid Metabolism in WAT

To examine the anti-obesity effects of *L. plantarum* ATG-K2, the expression of genes associated with lipid metabolism in WAT was investigated. The mRNA expression levels of lipogenesis-related genes, including PPARγ and FAS, are higher in the HFD group than in the NFD group. In contrast, the mRNA expression levels of PPARγ, SREPB1c, FAS, and DGAT1 are markedly downregulated in the 10 × 10^9^ CFU/day of *L. plantarum* ATG-K2 and Orlistat groups (Figure 3A). The mRNA expression levels of fatty acid oxidation-related genes, including PPARα and CPT1, and WAT browning-related genes, UCP1, PGC1α, Dio2, and ND5, are downregulated in the HFD group compared with the NFD group. These genes, PPARα and CPT1, UCP1, Prdm16, and ND5, are upregulated following treatment with 10 × 10^9^ CFU/day of *L. plantarum* ATG-K2 and in the Orlistat groups compared with the HFD group (Figure 3B,C).

### 2.8. Effects of L. plantarum ATG-K2 on Inflammation Markers in the Intestine and Serum

To determine the effects of *L. plantarum* ATG-K2 on inflammation, cytokine expression was measured in the intestine and serum. As shown in Figure 4A, there was no difference in mRNA expression levels of TNF-α and IL-6 in the intestine between NFD and HFD groups. Treatment with *L. plantarum* ATG-K2 and Orlistat significantly decreased TNF-α mRNA expression; however, IL-6 mRNA expression did not change. Moreover, these results are similar to the cytokine results in the serum of the 10 × 10^9^ CFU/day of *L. plantarum* ATG-K2 and Orlistat treatment groups (Figure 4B).

### 2.9. Changes of Gut Microbiota by L. plantarum ATG-K2

To examine the effects of *L. plantarum* ATG-K2 on the gut microbiota, the bacterial community from cecal samples of each experimental group was analyzed. The results of the family level relative abundance analysis are shown in Figure 5A. The family *Lactobacillaceae* showed a significant increase in the *L. plantarum* ATG-K2 and NFD groups. The family *Akkermansiaceae* exhibited a significant increase in the Orlistat group, and the *L. plantarum* ATG-K2 and NFD groups showed no change. The families *Bacteroidaceae* and *Helicobacteraceae*, showed a significant increase in the HFD group compared with the NFD group, but was not changed in the *L. plantarum* ATG-K2 and Orlistat groups. The family *Rikenellaceae* showed a significant decrease in the Orlistat groups. The family *Peptococcaceae* showed a significant decrease in the NFD and the Orlistat groups compared with the HFD group, but not in the *L. plantarum* ATG-K2 group (Figure 5B). Furthermore, gut bacteria richness was evaluated using chao1, and the diversity was evaluated using the Shannon index and Simpson’s index (Appendix A). No significant difference was observed in richness and diversity. Beta diversity was examined by principal coordinate analysis (PCoA) to compare the microbiota composition using UniFrac distance. Weighted UniFrac takes into account the relative abundance of species/taxa shared between samples. The microbiota from the NFD group was separated from those of the other groups. The *L. plantarum* ATG-K2 group exhibited similar changes to the Orlistat group relative to the HFD group (Figure 5C). Upon close examination of the PCo1 and PCo2 axis of our samples, we observed a significant difference in the NFD and Orlistat groups in PCo1 compared with the HFD group; while in PCoA2, a significant difference was observed in the *L. plantarum* ATG-K2 and the Orlistat groups compared with the HFD group (Appendix A).

### 2.10. Effects of L. plantarum ATG-K2 on SCFA Levels in Feces

Total short-chain fatty acids (SCFAs), acetic acid, butyric acid, propionic acid, and valeric acid levels in feces are lower in the HFD group than in the NFD group (Figure 6). A significant increase in SCFAs, acetic acid, propionic acid, and valeric acid levels was observed in the feces of both the *L. plantarum* ATG-K2 and Orlistat groups. Butyric acid levels were increased in the Orlistat groups. In addition, BSCFAs, iso-butyric acid, and iso-valeric acid levels in the feces of *L. plantarum* ATG-K2- and Orlistat-treated groups were significantly higher than those in the HFD group. These results suggest that *L. plantarum* ATG-K2 leads to increased levels of various SCFAs and branched short-chain fatty acids (BSCFAs), which can benefit HFD-induced obese mice.

## 3. Discussion

In the present study, we investigated the anti-obesity effects of *L. plantarum* ATG-K2 in 3T3-L1 adipocytes and HFD-induced obese mice. We found that *L. plantarum* ATG-K2 reduced lipid accumulation in 3T3-L1 adipocytes and exhibited anti-obesity effects against HFD-induced obesity, as demonstrated by the decreased body weight gain and various WAT weights without changes in food intake. In addition, H&E analysis showed that *L. plantarum* ATG-K2 exhibited an increased number of smaller adipocytes and decreased average adipocyte size. In parallel, it decreased serum leptin levels that are positively related to increased fat mass, suggesting that the reduced serum leptin levels by *L. plantarum* ATG-K2 may reflect the reduction of fat mass and adipocyte size in WAT. In addition, food intake and energy intake were not changed by *L. plantarum* ATG-K2. However, FER was lower with *L. plantarum* ATG-K2, which means that weight gain is low even if eating is the same and that the anti-obesity effects of *L. plantarum* ATG-K2 are not related to appetite suppression. These results suggest that *L. plantarum* ATG-K2 has an anti-obesity effect associated with reduced WAT weight, independent of food intake. Furthermore, we observed that *L. plantarum* ATG-K2 lowered serum TG and T-CHO levels in HFD-induced obese mice, consistent with our previous study [21]. Fatty acids that secrete adipose tissue in the obese state enter the liver, which may cause fatty liver and dyslipidemia in the blood [24]. These results indicate that *L. plantarum* ATG-K2 ameliorates dyslipidemia, accompanied by reduced adiposity and obesity.

Adipocyte hypertrophy is characterized by excessive lipid accumulation, which is associated with a complex network of various factors involved in lipid metabolism, including lipogenesis and fatty acid oxidation [25]. Furthermore, it has recently been reported that an increase in brite/beige adipocytes in WAT exhibits thermogenic properties and is linked to maintaining energy balance and preventing obesity with increased energy expenditure [26,27]. As expected, *L. plantarum* ATG-K2 reduced the mRNA expression levels of adipogenesis- and lipogenesis-related genes, including PPARγ, SREBP-1c, and FAS, and DGAT1 in 3T3-L1 adipocytes and the WAT of HFD-induced obese mice. In addition, it increased AMPK phosphorylation in 3T3-L1 adipocytes, and mRNA expression levels of fatty acid oxidation and adipocyte browning-related genes, PPARα, CTP1, UCP1, Prdm16, and ND5, which correspond to the inhibition of fat expansion in adipocytes and adipose tissue. These results suggest that *L. plantarum* ATG-K2 may regulate lipid metabolism by decreasing adipogenesis and lipogenesis and increasing fatty acid oxidation, with enhanced adipocyte browning in WAT, leading to reduced adiposity and anti-obesity effects. Additionally, *L. plantarum* ATG-K2 tended to inhibit fat absorption in the intestine because fecal TG levels tended to increase in HFD-induced obese mice. It is known that intestinal lipid absorption is essential in developing obesity, as increased intestinal lipid absorption causes fat accumulation in adipose tissue. Lipid absorption inhibitors, such as Orlistat, decreased fat mass through increased fecal lipid excretion [28], and previous reports have demonstrated that probiotics show anti-obesity effects through the inhibition of fat absorption [29,30]. This finding may partially explain the anti-obesity effect of *L. plantarum* ATG-K2 by modulating lipid metabolism.

Recently, attention on inflammation in the intestine has been increasing since it has been revealed that HFD induces inflammation by altering the gut microbiome and increasing intestinal permeability. Intestinal inflammation is an early consequence of HFD and may induce obesity as a causative factor in the onset of obesity [31]. The present study found that *L. plantarum* ATG-K2 decreased TNF-α mRNA expression levels in the small intestine and serum TNF-α levels. This finding is supported by previous reports that *L. plantarum* ATG-K2 reduced the expression levels of inflammatory cytokines such as TNF-α and IL-6, and NF-κB in the small intestine of high-fat and high-fructose diet fed rats [22]. These results indicate that *L. plantarum* ATG-K2 relieves intestinal inflammation, contributing to its protective effect against HFD-induced obesity.

Compelling evidence suggests that the gut microbiome is crucial in obesity progression [32]. The beneficial effect of the gut microbiome on obesity may be attributable to the derived metabolites SCFAs and BSCFAs. SCFAs are generated through the colonic fermentation of dietary fibers and BSCFAs are generated by fermentation of branched amino acids by the gut microbiome, respectively. Their production shares the same multienzyme pathway; however, there is a difference in substrate initiation of the pathway. In addition, a side-product such as H_2_ is utilized by other species in cross-feeding to avoid the accumulation of H_2,_ which would inhibit the ability of primary fermenters [33,34]. It suggests that the gut microbiome contributes enzymes with cross-feeding, and the beneficial effect of the gut microbiome is favorable to SCFA and BSCFA production. In the present study, we observed that *L. plantarum* ATG-K2 increased the relative abundance of *Lactobacillaceae* in a dose-dependent manner, suggesting that *L. plantarum* ATG-K2 belongs to the family *Lactobacillaceae*. *L. plantarum* ATG-K2 increased the relative abundance of the family *Akkermansiaceae* and decreased the relative abundance of the families *Bacteroidaceae*, *Rikenellaceae*, *Helicobacteraceae*, and *Peptococcaceae*; however, this difference was not significant. *L. plantarum* ATG-K2 increased SCFA levels (total SCFAs, acetic acid, propionic acid, and valeric acid) and BSCFA levels (iso-butyric acid and iso-valeric acid) in feces. The major SCFAs reduce lipogenesis and increase fatty acid oxidation, thermogenesis, and adipocyte browning in adipose tissue, muscle, and liver by increasing the expression of PGC1α, UCP1, and AMPK activation, which appear to have anti-obesity effects. Moreover, BSCFAs inhibit de novo lipogenesis in adipocytes by competing with acetyl-CoA to synthesize fatty acids [18,35,36]. In the present study, as a lactic acid bacterium, *L. plantarum* ATG-K2 may be attributed to an increase in the relative abundance of the *Lactobacillacease* family. It has been reported that the *Lactobacillaceae* family can produce lactic acid, and lactic acid is used to produce the SCFAs, acetic acid and propionic acid [37,38]. This suggests that a change of abundance of the *Lactobacillaceae* family has a partially beneficial effect on the gut microbiome composition. On the other hand, the increase of the genus *Lactobacillus* and *pediococcus* among the *Lactobacillacae* family was observed, but the abundance of the genus *pediococcus* was near to zero, while the abundance of the *Lactobacillacae* family was increased in a dose dependent manner. It is suggested that administration of *L. plantarum* ATG-K2 may be attributed to an increase of the *Lactobacillaceae* family. However, it needs to be investigated whether *L. plantarum* ATG-K2 may be attributed to increased abundances of the *Lactobacillaceae* family with a specific primer to detect *L. plantarum* ATG-K2. In addition, *Akkermansiaceae* utilizes colonic mucus as a carbohydrate source and is associated with increased pathogen susceptibility through enhanced bacterial colonization [39]; however, the *Akkermansiaceae* family might help the homeostasis of lipid levels and adipokines [40,41]. The function remains unknown. Therefore, we could not explain the repercussions of the families *Rikenellaceae* and *Peptococcaceae*; however, they were increased by HFD in the present study and other studies [39,42,43], and their relative abundance was affected by *L. plantarum* ATG-K2. Although more studies are warranted to identify active substances from *L. plantarum* ATG-K2, it is intriguing that *L. plantarum* ATG-K2 resulted in changes in the bacterial community, which may be associated with the alleviation of the gut microbiome dysbiosis induced by HFD. Consequently, gut microbiome modulation by *L. plantarum* ATG-K2, increasing SCFAs and BSCFAs, may serve as potential components to alleviate obesity.

*L. plantarum* ATG-K2 exerted anti-obesity effects by regulating lipid metabolism in 3T3-L1 adipocytes and HFD-induced obese mice. Although *L. plantarum* ATG-K2 lysates regulated lipid metabolism in 3T3-L1 adipocytes, it is possible that the combined action of modulation of the gut microbiome and its metabolites can contribute to ameliorating obesity by decreasing inflammation and lipid accumulation by regulating the expression of some crucial genes involved in these processes. Therefore, the underlying mechanisms of the anti-obesity effects of *L. plantarum* ATG-K2 may be proposed as follows: *L. plantarum* ATG-K2 restored the gut microbiome by increasing the abundance of beneficial bacteria and decreasing the bacterial abundance in HFD-induced obese mice. This in turn raised SCFA levels, resulting in a decreased inflammatory response and regulation of lipid metabolism-related gene expression, further improving obesity and lipid disorders.

## 4. Materials and Methods

### 4.1. Preparation of L. plantarum ATG-K2

*L. plantarum* ATG-K2 was prepared as previously described [20,21]. Briefly, *L. plantarum* ATG-K2 isolated from Korean fermented cabbage (Korean Collection for Type Culture [KCTC 13577BP]) was incubated in De Man Rogosa Sharp broth (Difco Laboratories Inc., Franklin Lakes, NJ, USA) at 37 °C for 16 h, then cell pellets were obtained by centrifugation (3000× *g*, 10 min, 4 °C), and washed three times with phosphate-buffered saline (PBS; pH 7.4, Welgene, Gyeongsan, Korea). To prepare the L. *plantarum* ATG-K2 lysates for in vitro experiments, the cell pellets were concentrated to 10× by resuspending in PBS and were treated with 10 mg/mL lysozyme (Sigma-Aldrich, St. Louis, MO, USA) at 37 °C for 2 h, and then were lysed with sonication. The resulting lysate (0.1 g) was used to measure the dry mass. Dry mass was determined by using a moisture analyzer (A&D Co., Ltd., Tokyo, Japan) that combines drying and on-line weight. The remaining resulting lysate mass was adjusted to 50 mg/mL to make a final stock solution concentration, and this was used for in vitro experiments. For in vivo experiments, the cell pellets were resuspended in cryoprotectant solution (Carboxymethyl cellulose, Changshu Wealthy Science and Technology Co., Ltd., Changshu, China; Threhalose, Hayashibara Co., Ltd., Tokyo, Japan; Skim milk, California Dairies Inc., Visalia, CA, USA) and lyophilized using an FD8508 freeze-dryer (ilShinBioBase, Dongduchen, Korea). Based on the survival rate after freeze drying, the administration dose in vivo was calculated. To determine the survival rate, the freeze-dried *L. plantarum* ATG-K2 powder was resuspended in 0.9% sterilized saline and prepared daily for animal experimental periods. To analyze, the freeze-dried *L. plantarum* ATG-K2 powder was suspended in saline, and this was serially diluted to 10^−8^. After a serial dilution, diluted *L. plantarum* ATG-K2 powder was inoculated onto MRS agar and incubated in 37 °C for 24 h. The colony number was counted, and the colony forming unit (CFU) was calculated by colony number x dilution factor.

### 4.2. Cell Culture

We purchased 3T3-L1 preadipocytes from the American Type Culture Collection (ATCC, Manassas, VA, USA). Dulbecco’s modified Eagle’s medium (DMEM) and Dulbecco’s PBS were purchased from Corning (Corning, NY, USA). Bovine calf serum (BCS), fetal bovine serum (FBS), antibiotic-antimycotic (Anti-Anti), 0.5% trypsin-ethylenediaminetetraacetic acid, and insulin were obtained from Gibco BRL (Rockville, NY, USA). Dexamethasone and 3-Isobutyl-1-methylxanthine (IBMX) were purchased from Sigma-Aldrich (St. Louis, MO, USA). The 3T3-L1 cells were cultured in 10% BCS-DMEM with 1% Anti-Anti at 37 °C in a humidified 5% CO_2_ incubator.

### 4.3. Cell Viability Assay

Cell viability assays were performed using a Cell Counting Kit-8 (CCK-8, Dojindo Laboratories, Kumamoto, Japan) according to the manufacturer’s protocol. The 3T3-L1 cells were plated in 96-well plates at a concentration of 1 × 10^4^ cells/well. After 24 h, the adherent cells were treated with lysate of *L. plantarum* ATG-K2 (25, 50, 100, 200, and 400 μg/mL) and incubated for 72 h. CCK-8 solution (10 μL) was added, and the cells were further incubated for 2 h. The absorbance was measured at 450 nm using an Epoch Microplate Spectrophotometer (BioTek Inc., Winooski, VT, USA).

### 4.4. Cell Differentiation

For adipocyte differentiation using 3T3-L1 adipocytes, 1 × 10^5^ cells/well were seeded into six-well plates and maintained at full confluence for two days. Cells were induced with differentiation medium (MDI: 10% FBS-DMEM with 1% Anti-Anti, 0.5 mM IBMX, 1 μM dexamethasone, and 10 μg/mL insulin, day 0). On day 3, the medium was replaced with an insulin medium (10% FBS-DMEM with 1% Anti-Anti, and 10 μg/mL insulin), then the medium was replaced with a growth medium (10% FBS-DMEM with 1% Anti-Anti, day 5) until day 7. Cells were treated with either PBS or *L. plantarum* ATG-K2 on days 0–7. A schematic diagram of adipocyte differentiation is shown in Figure 1B.

### 4.5. Oil Red O Staining

On day 7, differentiated 3T3-L1 cells were washed and fixed with 4% paraformaldehyde for 1 h. Fixed cells were washed with 60% isopropanol and allowed to dry. Dried cells were stained with a filtered Oil Red O working solution for 30 min. After washing four times with distilled water, the stained cells were photographed using an ECLIPSE Ts2 microscope (Nikon, Tokyo, Japan). To quantify staining, the stained lipid droplets were dissolved with 100% isopropanol, and the absorbance was measured at 500 nm using an Epoch Microplate Spectrophotometer (BioTek Inc., Winooski, VT, USA).

### 4.6. Animal Experiments

Five-week-old male C67BL/6J mice were purchased from Daehan BioLink Ltd. (Eumseong, Korea). The mice were housed under environmental conditions with a temperature of 22 ± 2 °C, 55 ± 5% humidity, and a 12 h light/dark cycle and were given free access to food and water. After 1 week of acclimatization, mice were divided into the following groups (n = 10/group): (1) NFD (4.5 kcal% Fat)-vehicle-treated group (NFD), (2) HFD (60 kcal% Fat, D12492, Research diet, New Brunswick, NJ, USA)-vehicle-treated group (HFD), (3) HFD + 4 × 10^9^ CFU of ATG-K2-treated group (HFD- 4 × 10^9^ ATG-K2), (4) HFD + 10 × 10^9^ CFU of ATG-K2 (HFD- 10 × 10^9^ ATG-K2), and (5) HFD + Orlistat (15.6 mg/kg orlistat) treated-group (HFD-Orlistat). For *L. plantarum* ATG-K2 and Orlistat groups, test samples were suspended in orally administered PBS for 8 weeks, and the NFD and HFD groups were treated by oral gavage with PBS. Bodyweight and food intake were measured weekly. The food efficiency rate was calculated as (total weight gain/total food intake) × 100.

### 4.7. Blood, WAT, Cecum, and Feces Collection

At the end of the experiment, the mice were sacrificed under anesthesia after fasting. Collected blood samples were centrifuged, and separated serum samples were stored at −80 °C until analysis. Epi, Peri, and Mes WAT samples were immediately excised, rinsed, weighed, frozen in liquid nitrogen and stored at –80 °C until analysis. The cecum and feces were collected in sterile tubes and stored at –80 °C until analysis.

### 4.8. Histological Analysis of WATs

Epi WAT was fixed with 10% formalin solution, embedded in paraffin, cut into 4 μm sections, and stained with H&E. Adipocyte size was observed under a light microscope (Olympus BX51, Olympus Optical Co., Tokyo, Japan) and determined by measuring the area taken up by 20 adipocytes in a stained section, and adipocyte cell number was measured using Image J1.49 software.

### 4.9. Serum Biochemical Parameter Analysis

Serum TG, T-CHO, LDL-CHO, HDL-CHO, and glucose levels were analyzed using a Hitachi-7020 automatic biochemical analyzer (Hitachi Medical, Tokyo, Japan). Serum leptin and adiponectin levels were measured by immunoassay using ELISA kits (Mouse Leptin and Adiponectin/Acrp30; R&D Systems, Minneapolis, MN, USA) according to the manufacturer’s protocols.

### 4.10. Fecal TG Analysis

Feces (0.1 g) were homogenized with 95% ethanol and centrifuged. The supernatant was mixed with Triton X-100 and sodium chlorate, and the fecal TG levels were measured using a commercial TG-S assay kit (Asan Pharm Co. Ltd., Seoul, Korea).

### 4.11. Quantitative Reverse-Transcription Polymerase Chain Reaction (RT-qPCR)

Total RNA was isolated using the easy-spin™ Total RNA Extraction Kit (iNtRON Biotechnology, Seongnam, Korea), and total RNA from Epi WAT was isolated with an RNeasy Lipid Tissue Mini Kit (QIAGEN, Hilden, Germany). cDNA was synthesized using GoScript™ Reverse Transcriptase (Promega, Madison, WI, USA). RT-qPCR was performed using TB Green^®^ Premix Ex Taq™ II (Takara Bio Inc., Shiga, Japan) on an ABI QuantStudio 3 (Applied Biosystems, Foster City, CA, USA). The primer sequences are listed in Appendix A.

### 4.12. Western Blot Analysis

The crude extracts from 3T3-L1 adipocytes were prepared using the lysis solution (PRO-PREP^TM^ with the halt^TM^ phosphatase inhibitor cocktail, iNtRON Biotechnology, and Thermo Fisher Scientific, Waltham, MA, USA, respectively). Proteins were quantified using a PRO-MEASURE protein measurement solution (iNtRON Biotechnology). Samples were separated with 6–10% sodium dodecyl sulfate-polyacrylamide gel electrophoresis and transferred to a polyvinylidene difluoride (PVDF) membrane. The membrane was blocked with skim milk for 1 h and then probed with the indicated antibodies. Antibodies against PPARγ, C/EBPα, AMPK, and p-AMPK (Thr172) were purchased from Cell Signaling Technology (Beverly, MA, USA). SREBP-1 and FAS were purchased from Santa Cruz Biotechnology (Dallas, TX, USA), and β-actin antibodies were purchased from Abcam (Cambridge, UK). Secondary antibodies were obtained from Promega. ImageQuant LAS 500 (GE Healthcare Life Sciences, Chicago, IL, USA) was used for visualization.

### 4.13. Cecal Microbiota Analysis

Genomic DNA was extracted from cecal samples using the QIAamp PowerFecal Pro DNA Kit (QIAGEN). The quantity and quality of extracted DNA were measured using a Qubit 3.0 Fluorometer (Thermo Fisher Scientific) and agarose gel electrophoresis, respectively. The V4 hypervariable regions of the bacterial 16S rRNA were amplified with unique 8 bp barcodes and sequenced on the Illumina MiSeq PE300 platform according to the standard protocol [44]. The raw sequence data were submitted to the NCBI SRA database (NCBI BioProject PRJNA744955). Raw reads were analyzed using the QIIME pipeline [45] with the SILVA 132 database [46]. The nonparametric Kruskal–Wallis test was used to compare the differences in diversity indices and microbial taxa. The weighted UniFrac distances were previously obtained and used for PCoA [47].

### 4.14. Analysis of Fecal SCFAs

#### 4.14.1. Chemicals

Acetic, butyric, propionic, valeric, iso-butyric, and iso-valeric acids of analytical grade were purchased from Sigma-Aldrich. Formic acid, ammonium formate, 3-nitrophenylhydrazine (3NPH)∙HCl (97%), N-(3-dimethylaminopropyl)-N′-ethylcarbodiimide (EDC)∙HCl, acetonitrile, methanol (MeOH), and water were purchased from Sigma-Aldrich. All solvents were of high-performance liquid chromatography grade.

#### 4.14.2. Sample Preparation

Sample preparation was achieved by homogenization and protein removal. Briefly, the raw fecal homogenate was prepared by using 50–200 mg of feces, adding 4 mL ice-cold MeOH with 1% formic acid per g of feces, and homogenizing in a Vibra-Cell Processors VCX 750 (Sonics & Materials, Inc., Newtown, CT, USA) for 30 s. After the fecal homogenate was thoroughly mixed by vortexing for 30 min, it was left for 2 h (4 °C) to enable solidification of the precipitate. After centrifugation at 18,000× *g* for 30 min at 4 °C, the supernatant was filtered using a PVDF 0.22 μm pore size syringe filter (Millipore, Burlington, MA, USA). For derivatization, 40 μL of the filtered sample was mixed with 20 μL of 200 mM 3NPH in 50% aqueous acetonitrile and 20 μL of 120 mM EDC solution. The mixture was allowed to react at 40 °C for 30 min. The derivatization sample was then injected into a liquid chromatography-tandem mass spectrometry (LC-MS/MS) system.

#### 4.14.3. LC-MS/MS Instrumentation and Analytical Conditions

LC-MS/MS analyses were carried out using an Exion LC system connected to a QTRAP 4500 mass spectrometer (AB SCIEX, Framingham, MA, USA). Separation was achieved with a Unison UK-C18 HT column (100 mm × 2 mm, 3 μm particle size; Imtakt, Kyoto, Japan) at a flow rate of 0.35 mL min^−1^ and column temperature of 40 °C. An injection volume of 1 μL was used for the autosampler injection. A gradient mobile phase consisting of water/formic acid (100:0.01, *v*/*v*; solvent A) and acetonitrile/formic acid (100:0.01, *v*/*v*; solvent B) was employed. The binary solvent elution gradient was optimized at 15% B for 2 min, 15–40% B for 14 min, 100% B for 2 min, and a final return to 15% B within 1 min. The column was equilibrated for 1 min at 15% B between injections. The total run time was 20 min. The analysis was carried out using an electrospray ionization source in negative mode. The operating conditions were as follows: ion spray voltage, 4500 V; curtain gas, 25 psi; collision gas, medium; ion source gas 1 and ion source gas 2, 50 and 50 psi; turbo spray temperature, 450 °C; entrance potential, −10 V; and collision cell exit potential, −9 V. Nitrogen was used in all cases. Analytes were quantified by multiple reaction monitoring using the following precursor to product ion transitions and parameters: acetic acid, *m*/*z* 193.9 → 152.0 with declustering potential (DP) −70 V and collision energy (CE) −20 eV; butyric acid, *m*/*z* 221.9 → 151.9 with DP −75 V and CE −22 eV; propionic acid, *m*/*z* 207.9 → 165.1 with DP −55 V and CE −20 eV; valeric acid, *m*/*z* 235.9 → 151.9 with DP −85 V and CE −24 eV; iso-butyric acid, *m*/*z* 221.6 → 179.0 with DP −95 V and CE −20 eV; iso-valeric acid, *m*/*z* 235.9 → 151.6 with DP −75 V and CE −24 eV. SCIEX OS 2.0.0 software was used for data acquisition and processing, while Analyst 3.3 software was used for data analysis.

### 4.15. Statistical Analysis

Data are presented as the mean ± SEM. Differences between groups were analyzed by one-way analysis of variance followed by Dunnett’s multiple comparison tests using Prism 8.0 software (GraphPad Software Inc., San Diego, CA, USA). Differences were considered statistically significant at *p* < 0.05.

## 5. Conclusions

In the present study, we found that *L. plantarum* ATG-K2 exhibited anti-obesity and anti-hyperlipidemic effects in 3T3-L1 adipocytes and HFD-induced obese mice, which was accompanied by an inhibition of lipid accumulation and the inflammatory response, and regulation of the expression of lipid metabolism and adipocyte browning-related genes. Specifically, *L. plantarum* ATG-K2 restored the composition of the gut microbiota in the cecum, resulting in an increase in SCFAs, which may be attributed to gene expression regulation associated with lipid metabolism and inflammation to inhibit lipid accumulation. Taken together, these findings suggest that *L. plantarum* ATG-K2 has a beneficial effect in ameliorating obesity and offers novel therapeutic strategies for obesity.

## Figures and Tables

**Figure 1 ijms-22-12665-f001:**
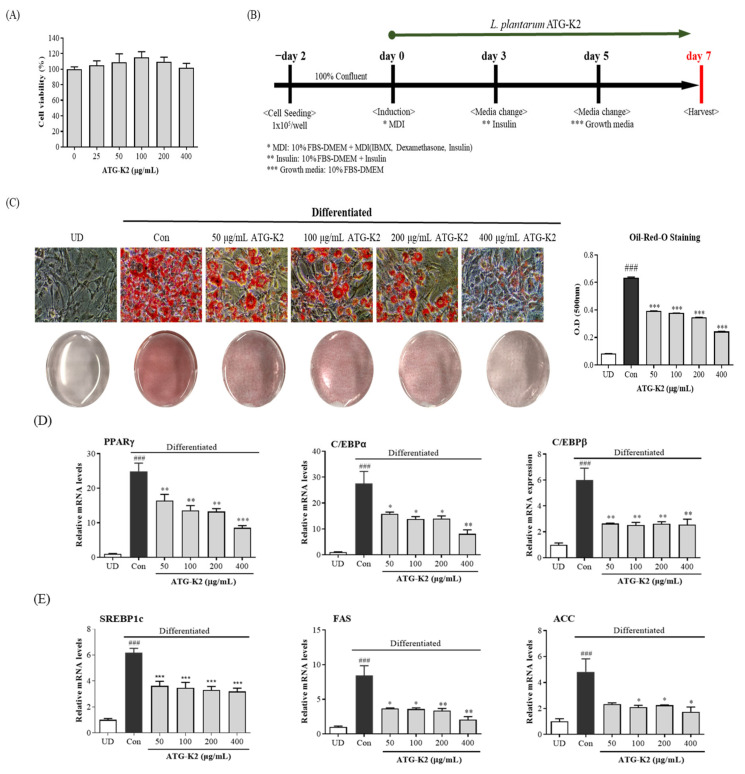
Effects of *L. plantarum* ATG-K2 on lipid accumulation in 3T3-L1 cells. (**A**) Cell viability, (**B**) schedule of the culture with lysate of *L. plantarum* ATG-K2 during the adipogenic differentiation of 3T3-L1 cells, (**C**) measurement of lipid droplets by Oil Red O staining (scale bar: 50 μm), the mRNA expression level of (**D**) adipogenic and (**E**) lipogenesis-related genes, (**F**) protein expression levels related to adipogenesis and lipogenesis, and (**G**) level of AMPK phosphorylation. UD: Undifferentiated; Con: Vehicle control; 50~400 μg/mL ATG-K2: 50~400 μg/mL of *L. plantarum* ATG-K2. All values are expressed as mean ± SEM. ^###^
*p* < 0.005 vs. UD; * *p* < 0.05, ** *p* < 0.01, and *** *p* < 0.001 vs. Con.

**Figure 2 ijms-22-12665-f002:**
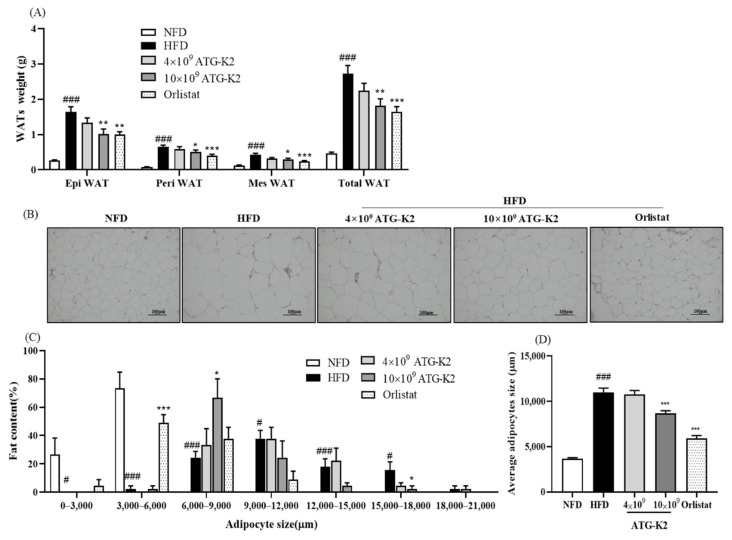
Effect of *L. plantarum* ATG-K2 on WAT weight and adipocyte sizes in HFD-induced obese mice. (**A**) WAT weight, (**B**) hematoxylin and eosin (H&E) stain of Epi WAT, (**C**) adipocyte cell size distribution, and (**D**) average adipocytes. NFD: normal fat diet; HFD: high-fat diet; HFD-4 × 10^9^~10 × 10^9^ ATG-K2: HFD + 4 × 10^9^~10 × 10^9^ CFU/day *L. plantarum* ATG-K2; HFD-Orlistat: HFD + Orlistat 15.6 mg/kg. Data are expressed as mean ± SEM (n = 10). ^#^
*p* < 0.05 and ^###^
*p* < 0.005 vs. NFD group; * *p* < 0.05, ** *p* < 0.01 and *** *p* < 0.005 vs. HFD group.

**Figure 3 ijms-22-12665-f003:**
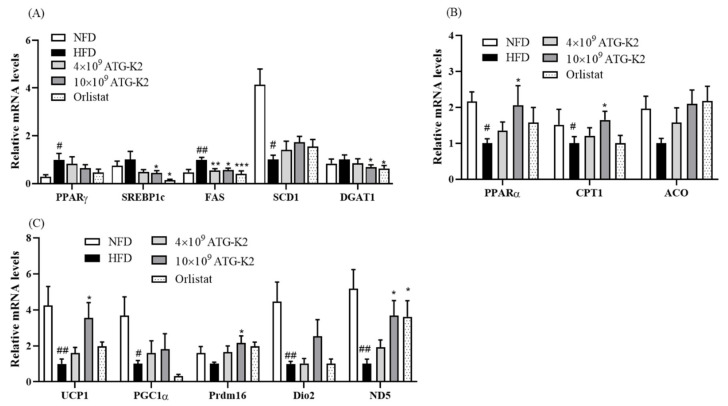
Effect of *L. plantarum* ATG-K2 on expression levels of genes associated with lipid metabolism in Epi WAT of the HFD-induced obese mice. (**A**) mRNA expression levels of lipogenesis-related genes, (**B**) mRNA expression levels of fatty acid oxidation-related genes, and (**C**) mRNA expression levels of WAT browning-related genes. NFD: normal fat diet; HFD: high-fat diet; HFD-4 × 10^9^~10 × 10^9^ ATG-K2: HFD + 4 × 10^9^~10 × 10^9^ CFU/day *L. plantarum* ATG-K2; HFD-Orlistat: HFD + Orlistat 15.6 mg/kg. Data are expressed as mean ± SEM (n = 10). ^#^
*p* < 0.05 and ^##^
*p* < 0.01 vs. NFD group; * *p* < 0.05, ** *p* < 0.01, and *** *p* < 0.005 vs. HFD group.

**Figure 4 ijms-22-12665-f004:**
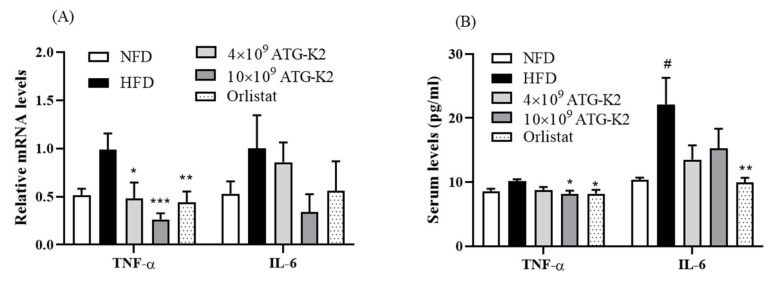
Effect of *L. plantarum* ATG-K2 on inflammatory cytokines in the serum and intestine of HFD-induced obese mice. (**A**) Cytokine mRNA expression levels in intestine and (**B**) cytokine levels in serum. NFD: normal fat diet; HFD: high-fat diet; HFD-4 × 10^9^~10 × 10^9^ ATG-K2: HFD + 4 × 10^9^~10 × 10^9^ CFU/day *L. plantarum* ATG-K2; HFD-Orlistat: HFD + Orlistat 15.6 mg/kg. Data are expressed as mean ± SEM (n = 10). ^#^ *p* < 0.05 vs. NFD group; * *p* < 0.05, ** *p* < 0.01, and *** *p* < 0.005 vs. HFD group.

**Figure 5 ijms-22-12665-f005:**
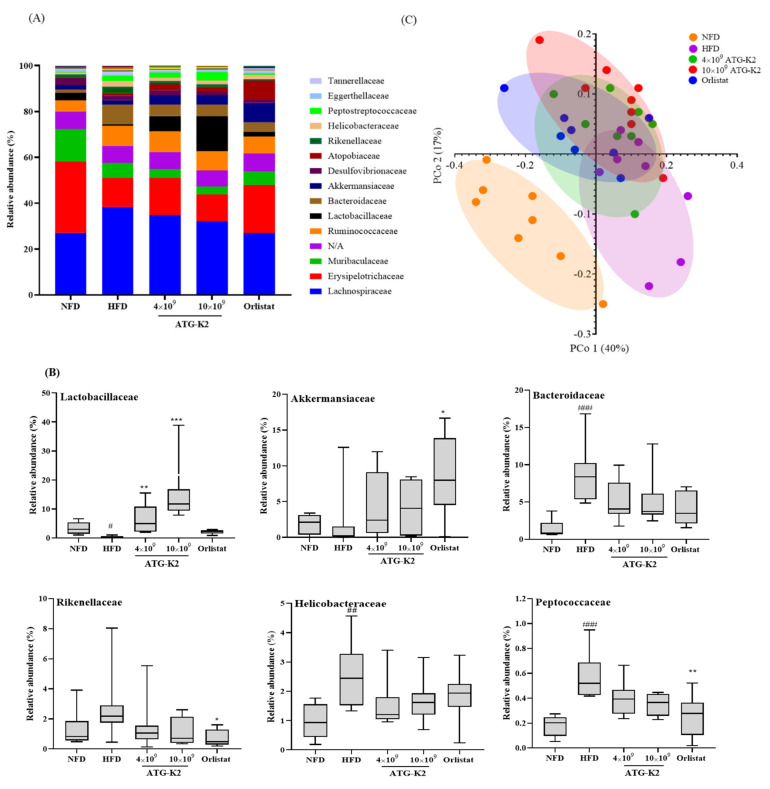
Changes in the cecal bacterial community of *L. plantarum* ATG-K2 in HFD-induced obese mice. (**A**) Average relative abundance of the family level in each group, (**B**) relative abundance of the family, and (**C**) principal coordinate analysis plots. NFD: normal fat diet; HFD: high-fat diet; HFD-4 × 10^9^~10 × 10^9^ ATG-K2: HFD + 4 × 10^9^~10 × 10^9^ CFU/day *L. plantarum* ATG-K2; HFD-Orlistat: HFD + Orlistat 15.6 mg/kg. Data are expressed as mean ± SEM (*n* = 10). ^#^
*p* < 0.05, ^##^
*p* < 0.01, and ^###^
*p* < 0.005 vs. NFD group; * *p* < 0.05, ** *p* < 0.01, and *** *p* < 0.005 vs. HFD group.

**Figure 6 ijms-22-12665-f006:**
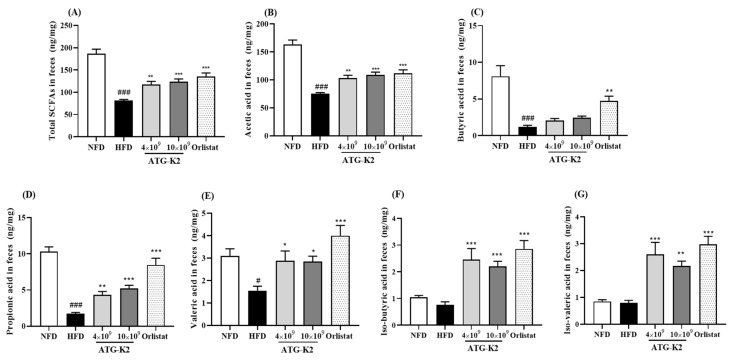
Effects of *L. plantarum* ATG-K2 on fecal SCFA and BSCFA levels. (**A**) Total SCFAs, (**B**) acetic acid, (**C**) butyric acid levels, (**D**) propionic acid levels, (**E**) valeric acid in feces, (**F**) iso-butyric acid, and (**G**) iso-valeric acid levels in feces. NFD: normal fat diet; HFD: high-fat diet; HFD-4 × 10^9^~10 × 10^9^ ATG-K2: HFD + 4 × 10^9^~10 × 10^9^ CFU/day *L. plantarum* ATG-K2; HFD-Orlistat: HFD + Orlistat 15.6 mg/kg. Data are expressed as mean ± SEM (*n* = 10). ^#^
*p* < 0.05 and ^###^
*p* < 0.005 vs. NFD group; * *p* < 0.05, ** *p* < 0.01, and *** *p* < 0.005 vs. HFD group.

**Table 1 ijms-22-12665-t001:** Effects of *L. plantarum* ATG-K2 on body weight and food intake in HFD-induced obese mice.

	NFD	HFD	HFD		
			4 × 10^9^ ATG-K2	10 × 10^9^ ATG-K2	Orlistat
Initial body weight (g)	19.71 ± 1.24	22.01 ± 0.56	22.60 ± 0.34	22.42 ± 0.42	22.38 ± 0.48
Final body weight (g)	25.02 ± 0.51	34.28 ± 1.06 ^###^	32.70 ± 0.93	31.06 ± 0.94 **	29.23 ± 0.88 ***
Body weight gain (g)	5.31 ± 0.64	12.33 ± 1.25 ^###^	10.10 ± 0.74	8.10 ± 0.9 *	6.85 ± 0.96 ***
Food intake (g/mouse/day)	2.86 ± 0.06	2.44 ± 0.08 ^##^	2.69 ± 0.11	2.40 ± 0.06	2.78 ± 0.06 *
Energy intake (kcal/mouse/day)	11.01 ± 0.23	12.79 ± 4.2 ^##^	14.10 ± 0.58	12.57 ± 0.31	14.57 ± 0.31 *
FER ^a^	4.04 ± 0.49	10.99 ± 1.11 ^###^	8.16 ± 0.59	7.35 ± 0.88 *	5.35 ± 0.75 ***

^a^ FER (food efficiency ratio) = (total weight gain/total food intake) × 100. NFD: normal fat diet; HFD: high-fat diet; HFD-4 × 10^9^~10 × 10^9^ ATG-K2: HFD + 4 × 10^9^~10 × 10^9^ CFU/day *L. plantarum* ATG-K2; HFD-Orlistat: HFD + Orlistat 15.6 mg/kg. Data are expressed as mean ± SEM (n = 10). ^##^
*p* < 0.01 and ^###^
*p* < 0.005 vs. the NFD group; * *p* < 0.05, ** *p* < 0.01, and *** *p* < 0.005 vs. the HFD group.

**Table 2 ijms-22-12665-t002:** Effects of *L. plantarum* ATG-K2 on serum biochemical parameters and fecal TG in HFD-induced obese mice.

	NFD	HFD	HFD		
			4 × 10^9^ ATG-K2	10 × 10^9^ ATG-K2	Orlistat
TG (mg/dL)	128.17 ± 2.41	145.00 ± 13.02	109.80 ± 11.29	106.44 ± 4.07 *	147.60 ± 11.16
T-CHO (mg/dL)	109.33 ± 2.54	157.64 ± 7.57 ^###^	141.10 ± 6.53	135.67 ± 3.99 *	134.25 ± 4.77 *
HDL-CHO (mg/dL)	66.78 ± 0.83	78.00 ± 1.12 ^##^	73.80 ± 2.97	75.70 ± 2.52	76.67 ± 1.48
LDL-CHO (mg/dL)	10.78 ± 0.40	12.09 ± 0.59	12.20 ± 0.89	11.60 ± 0.60	10.25 ± 0.37
Glucose (mg/dL)	34.22 ± 3.81	148.45 ± 13.11 ^###^	130.30 ± 11.87	124.00 ± 13.55	118.67 ± 5.83
Adiponectin (μg/mL)	11.78 ± 0.53	10.56 ± 0.88 ^#^	11.36 ± 0.37	10.89 ± 0.26	10.93 ± 0.27
Leptin (μg/mL)	1.21 ± 0.3	25.28 ± 3.88 ^###^	19.77 ± 3.40	14.06 ± 2.22 *	9.17 ± 1.11 ***
Fecal TG (mg/dL)	177.29 ± 17.53	269.75 ± 36.84	367.20 ± 23.47	356.65 ± 35.95	484.10 ± 78.62 *

NFD: normal fat diet; HFD: high-fat diet; HFD-4 × 10^9^~10 × 10^9^ ATG-K2: HFD + 4 × 10^9^ ~ 10 × 10^9^ CFU/day *L. plantarum* ATG-K2; HFD-Orlistat: HFD + Orlistat 15.6 mg/kg. TG: triglyceride; T-CHO: Total cholesterol; HDL-CHO: high-density lipoprotein cholesterol; LDL-CHO: low-density lipoprotein cholesterol. Data are expressed as mean ± SEM (*n* = 10). ^#^
*p* < 0.05, ^##^
*p* < 0.01, and ^###^
*p* < 0.005 vs. the NFD group; * *p* < 0.05 and *** *p* < 0.005 vs. the HFD group.

## Data Availability

Not applicable.

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
