# Peer review of "Lactiplantibacillusplantarum ATG-K2 Exerts an Anti-Obesity Effect in High-Fat Diet-Induced Obese Mice by Modulating the Gut Microbiome"

_ijms, 2021, doi:10.3390/ijms222312665_

Round 1
Reviewer 1 Report
YS Lee et al. reported that Lactobacillus plantarum ATG-K2 exerts an anti-obesity effect in high-fat-fed obese mice by modulating the intestinal microflora. Although the results are interesting for the most part, minor revisions are needed to adopt this report in the IJMS.
In HFD, the PCOA plot was shifted in the ATG and Xen groups compared to the HFD alone group.
In HFD, Figure 5C shows a shift in the PCOA plot in the ATG and Xen groups compared to the HFD alone group. In the supple fig, both chao1 and shannon were significantly decreased in the Xen group, although the difference was not significant. However, it would be good to add a rational explanation for the fact that the beta diversity seems to be rather close to NFD.
In Table 2, the authors show a decrease in total cholesterol in the ATG and Xen groups, but there is almost no decrease in LDL cholesterol in the ATG group. Can you provide a clear explanation for the cholesterol fractions?
Author Response
Comments and Suggestions for Authors
YS Lee et al. reported that Lactobacillus plantarum ATG-K2 exerts an anti-obesity effect in high-fat-fed obese mice by modulating the intestinal microflora. Although the results are interesting for the most part, minor revisions are needed to adopt this report in the IJMS.
In HFD, the PCOA plot was shifted in the ATG and Xen groups compared to the HFD alone group. In HFD, Figure 5C shows a shift in the PCOA plot in the ATG and Xen groups compared to the HFD alone group. In the supple fig, both chao1 and shannon were significantly decreased in the Xen group, although the difference was not significant. However, it would be good to add a rational explanation for the fact that the beta diversity seems to be rather close to NFD.
- Responses
- Thank you for your comment. We added an explanation as following. In the present study, L. plantarum ATG-K2 was increased the abundance of the Lactobacillaceae family. Lactobacillacease family can produce a lactic acid, and lactic acid is used to produce the SCFAs. This suggests that change of abundance of Lactobacillaceae family has partially beneficial effect on gut microbiome composition.
In Table 2, the authors show a decrease in total cholesterol in the ATG and Xen groups, but there is almost no decrease in LDL cholesterol in the ATG group. Can you provide a clear explanation for the cholesterol fractions?
Thank you for your comments.
- Response Cholesterol is divided into Chylomicron, very-low-density lipoprotein (VLDL), intermediate-density lipoprotein (IDL), LDL and HDL, based on the major cholesterol-carrying lipoproteins. LDL-CHO is produced after remodeling in the plasma and the liver, and accomplish cholesterol transport to cells requiring lipids. Since T-CHO include the LDL and HDL as well as VLDL and Chylomicron, there may be no change LDL. Indeed, the liver is one of two crucial organs for cholesterol homeostasis and involved in high cholesterol biosynthesis. It needs to investigate the effect on cholesterol metabolism including lipoproteins, lipoprotein lipases and LDL-R in liver.
Reviewer 2 Report
The article of Lee and col. describes in vitro and in vivo anti-obesity effect of a probiotic strain. The article is well written and can be a contribution for the field; however I have several major points that need attention which are mainly related with interpretation of the data, discussion of the results and conclusions. Authors can not interpret a tendency as a positive or negative results. This point must be corrected and discussion of results that show a tendency must be eliminated. Other important issue is related with the rationale to use the concentrations of probiotic in the in vitro studies. These concentrations should not be expressed in weight/volume, but by CFU/g of lyophilized product. Please explain if these concentrations of probiotic can reach adipose tissue/cells. In addition, authors must revise the effect of HFD or probiotic on the gut microbiota, because alpha diversity was not affected and only some bacterial families modified their relative abundances. I did not consider this effect as a dysbiosis or a modulation of the microbiota by the probiotic strain. Authors need to discuss their results more critically. Several other comments are detailed below. In brief, modifications in all sections are needed, especially in abstract, results, discussion and conclusion sections, after the description of the results will be corrected.
Major comments:
Why a dietary intervention is not proposed as a measure to prevent obesity?
L52-53 Please check the meaning of this sentence. What is a “normalization of the gut microbiome”?
L53-54 Please describe which dysbiosis or imbalance of the microbiota is associated with obesity including relevant references.
L57-59 The modulation of the microbiota by probiotics is performed by some probiotic strains. Indeed, this effect on microbiota is not included in the definition of probiotics. Please do not generalize this effect for all probiotics.
It is important to know if the lyophilized probiotic contains live bacteria and to determine the concentration of the live probiotic per gram of lyophilized product (CFU/g). What is the rationale to test these concentrations to test 3T3-L1 cell viability? What is the probability that these probiotic concentrations reach these cells in in vivo conditions? The same comment for lipid accumulation, expression, AMPK activation analysis.
Please include energy consumed by each group of animals that can explain the higher weight gain in HFD group, despite a lower food intake.
L166-167 “Fecal TG showed a tendency to increase in both the L. plantarum ATG-K2 groups” If you did not observed a statistically difference, the level of TG is the same. Then remove that L. plantarum ATG-K2 increased fecal TG levels in the abstract.
L176-178 and Figure 3A The SREPB1c and DGAT1 expression is not higher in HFD group compared to NFD.
L178 -179 both concentrations of probiotic did not exert the same effect on gene expression, and not all gene were downregulated, please revise
L179-183 and Figure 3 B and C. these genes are upregulated compared to which group? HFD? ACO gene expression was not affected.
L193-195 eliminate “tend to be higher”, this means that the expression level was the same or not affected.
L212-213 eliminate the results related with a “tendency”, this is not an effect
L213-215 The same comments for this sentence.
L217-219 the same comments for these results
Please check the Supp Fig 1, the panel do not correspond with the legend.
L222-225 An statistical analysis must be performed to evaluate effect of the group in microbiota.
L235-236 butyric acid is not lower in HFD compared to NFD
L238-239 eliminate the results related with a “tendency”, this is not an effect
How the beneficial effect of the gut microbiome on obesity may be attributable to the derived metabolites SCFAs and BSCFAs?
Can we consider a dysbiosis the modification of the relative abundances of 4 bacterial families?
The authors mentioned that the probiotic can restore the gut microbiota by increasing the abundance of benefic bacteria. Which ones? The Lactobacillaceae family? Can this increase be attributed to the administration of the Lactobacillus strain? Did you check similarities between the probiotic and Lactobacillae families? Which genera increased their abundances?
Results indicate that microbiota identification was performed on cecal samples, but Methods indicate that the analysis was done in fecal samples. Please revise.
BSCFAs are the results of fermentation of the protein. Please discuss this point.
L309-314 please include the concentration of BSCFAs that induce these effects, and compare to those concentrations determined in fecal samples.
L282-289 The inhibition of fat absorption can be exerted by inhibiting the pancreatic lipase, such as orlistat, or at the intestinal level. Please specify which mechanisms are you discussing here.
L351-352 How the probiotic pellet was treated with lysozyme? Did you resuspend the pellet in PBS? What concentration of lysozyme did you used? What was the purpose to prepare a probiotic lysate if this was not used in experiments?
Minor comment:
Line 49 “spp.” should not be italized
L362 what is “anti-anti”
Please use the name of the chemical compound instead of the commercial brand Xenical.
Author Response
Comments and Suggestions for Authors
The article of Lee and col. describes in vitro and in vivo anti-obesity effect of a probiotic strain. The article is well written and can be a contribution for the field; however I have several major points that need attention which are mainly related with interpretation of the data, discussion of the results and conclusions. Authors can not interpret a tendency as a positive or negative results. This point must be corrected and discussion of results that show a tendency must be eliminated. Other important issue is related with the rationale to use the concentrations of probiotic in the in vitro studies. These concentrations should not be expressed in weight/volume, but by CFU/g of lyophilized product. Please explain if these concentrations of probiotic can reach adipose tissue/cells. In addition, authors must revise the effect of HFD or probiotic on the gut microbiota, because alpha diversity was not affected and only some bacterial families modified their relative abundances. I did not consider this effect as a dysbiosis or a modulation of the microbiota by the probiotic strain. Authors need to discuss their results more critically. Several other comments are detailed below. In brief, modifications in all sections are needed, especially in abstract, results, discussion and conclusion sections, after the description of the results will be corrected.
Major comments:
Why a dietary intervention is not proposed as a measure to prevent obesity?
- Response
Thank you for your comments. As your comment, dietary intervention can prevent obesity. However, this study firstly focused on the intrinsic effect of L. plantarum ATG-K2 itself on obesity. In future, it needs to investigate the combination effects of L. plantarum ATG-K2 and dietary intervention.
L52-53 Please check the meaning of this sentence. What is a “normalization of the gut microbiome”?
L53-54 Please describe which dysbiosis or imbalance of the microbiota is associated with obesity including relevant references.
L57-59 The modulation of the microbiota by probiotics is performed by some probiotic strains. Indeed, this effect on microbiota is not included in the definition of probiotics. Please do not generalize this effect for all probiotics.
- Response
Thank you for your comments. As your comments, it was replaced as follow (line 47-75).
Furthermore, the gut microbiome has recently been considered an essential regulator of host metabolism. Dysbiosis or imbalance of the gut microbiome is associated with obesity and its complications [8]. Dysbiosis or imbalanced gut microbiome means disruption of balance between the gut microbiota and host and reflects negative shifts in abundance, diversity, and relative distribution of the gut microbiome composition [5, Lozupone et al., 2012]. Dysbiosis induced by a high-fat diet induces lipopolysaccharides production from gram-negative bacteria in the gut and increases intestinal permeability, which causes metabolic endotoxemia. Consequently, metabolic endotoxemia induces low-grade chronic inflammation in obesity and its complications through a Toll-like receptor-mediated inflammatory pathway [ Cani et al., 2007]. We also observed the change in the Firmicutes/Bacteroidetes ratio, decreases of Bifidobacterium spp. and Akkermansia, and increase of populations of Ruminococcaceae and Rikenellaceae during high-fat diet-induced obese and metabolic disorders [Cani, 2013; Okeke, 2014]. The gut microbiome produces short-chain fatty acids (SCFAs), such as acetic, propionic, and butyric acids, which playcritical roles in preventing and treating obesity and its complications by improving glucose and lipid homeostasis and decreasing inflammation [13]. In parallel, extensive metagenome-wide studies found a reduction in butyrate-producing microbes together with an increase in opportunistic pathogens in obesity and type 2 diabetes mellitus of China and Europe [Vallianou et al., 2019]. On the other hand, some probiotics exert metabolically beneficial effects by modulating the gut microbiome and its metabolites [10-12]. For example, L. rhamnosus LS-8 and L. crustorum MN047 manipulated gut microbiota by decreasing the abundance of Bacteroides and Desulfovibrio and increasing Lactobacillus and Bifdobacterium, which led to increasing the levels of SCFAs in feces [7]. Therefore, targeting changes in the gut microbiome using probiotics has emerged as an attractive strategy.
Reference
- Lozupone, C.A.; Stombaugh, J.I.; Gordon, J.I.; Jansson, J.K.; Knight, R., Diversity, stability and resilience of the human gut microbiota. Nature 2012, 489, (7415), 261-273.
- Cani, P.D., Amar. J., Iglesias, M.A., Poggi, M., Knauf, C., Bastelica, D., Neyrinck, A.M., Fava, F., Tuohy, K.M., Chabo, C., Waget, A., Delmée, E., Cousin, B., Sulpice, T., Chamontin, B., Ferrières, J., Tanti, J.F., Gibson, G.R., Casteilla, L., Delzenne, N.M., Alessi, M.C., Burcelin, R., Metabolic endotoxemia initiates obesity and insulin resistance. Diabetes 2007, 56, 1761–1772.
- Cani, P.D., Gut microbiota and obesity: lessons from the microbiome. Brief Funct Genomics 2013, 12, (4), 381-387.
- Okeke, F., The role of the gut microbiome in the pathogenesis and treatment of obesity. Glob Adv Health Med. 2014, 3, (3), 44-57.
- Vallianou, N.; Stratigou, T.; Christodoulatos, G.S.; Dalamaga, M., Understanding the role of the gut microbiome and microbial metabolite in obesity and obesity-associated metabolic disorders: Current evidence and perspective. Curr Obes Rep. 2019,8,317-332.
It is important to know if the lyophilized probiotic contains live bacteria and to determine the concentration of the live probiotic per gram of lyophilized product (CFU/g). What is the rationale to test these concentrations to test 3T3-L1 cell viability? What is the probability that these probiotic concentrations reach these cells in in vivo conditions? The same comment for lipid accumulation, expression, AMPK activation analysis.
- Response
Thank you for your comments.
The survival rate of L. plantarum ATG-K2 after lyophilization is a about 70%. Based on the survival rate after freeze-drying, the administration dose in vivo was calculated. Lysates of L. plantarum ATG-K2 has been used in 3T3-L1 experiments. L. plantarum ATG-K2 was incubated in MRS broth, then cell pellets were obtained by centrifugation, and washed three times with phosphate-buffered saline (PBS; pH7.4). The cell pellets were treated lysozyme (Sigma-Aldrich, St. Louis, MO, USA) and then were lysed with sonication. (Refer to revised manuscript 4.1).
It has been known that probiotics exert metabolically beneficial effects by modulating the gut microbiome and its metabolites such as short chain fatty acids (SCFAs), and probiotics didn’t affect on metabolism in peripheral tissues in vivo. SCFAs play key roles in host metabolism [Canfora et al., 2019, He and Shi, 2017]. Also, gut microbiome modulation and its metabolite profiling (kinds, concentration) varies by probiotics species and strain specific effect and intestinal environment. So, it can’t calculate the probability that probiotic concentrations reach these cells in in vivo conditions.
Reference
- Canfora, E. E.; Meex, R. C. R.; Venema, K.; Blaak, E. E., Gut microbial metabolites in obesity, NAFLD and T2DM. Nat Rev Endocrinol 2019, 15, (5), 261-273.
- He, M.; Shi, B., Gut microbiota as a potential target of metabolic syndrome: the role of probiotics and prebiotics. Cell Biosci 2017, 7, 54.
Please include energy consumed by each group of animals that can explain the higher weight gain in HFD group, despite a lower food intake.
- Response
Thank you for your comments. As your comments, explanation of energy intake was added into Results and Discussion section, and Table 1 (line 158).
2.4. Effects of L. plantarum ATG-K2 on body weight and food intake rate (line 137~139)
Food intake and energy intake in the HFD group were lower than that in the NFD and Orlistat groups, however, the L. plantarum ATG-K2 group did not affect food intake and energy intake.
Discussion section (line 284-287)
In addition, food intake and energy intake were not changed by L. plantarum ATG-K2, however, FER was lower by L. plantarum ATG-K2, which means that weight gain is low even if eating same, and that the anti-obesity effects of L. plantarum ATG-K2 are not related to appetite suppression.
L166-167 “Fecal TG showed a tendency to increase in both the L. plantarum ATG-K2 groups” If you did not observed a statistically difference, the level of TG is the same. Then remove that L. plantarum ATG-K2 increased fecal TG levels in the abstract.
- Response
Thank you for your comments. As your comments, it was deleted in Abstract section and was replaced as follow (line 183~185)
Fecal TG showed no change in both the L. plantarum ATG-K2 groups and significantly increased in the Orlistat group.
L176-178 and Figure 3A The SREPB1c and DGAT1 expression is not higher in HFD group compared to NFD.
L178 -179 both concentrations of probiotic did not exert the same effect on gene expression, and not all gene were downregulated, please revise
L179-183 and Figure 3 B and C. these genes are upregulated compared to which group? HFD? ACO gene expression was not affected.
- Response
Thank you for your comments. As your comments, it was corrected as follow (line 194~203).
2.7. Effects of L. plantarum ATG-K2 on lipid metabolism in WAT
The mRNA expression levels of lipogenesis-related genes, including PPARγ and FAS, are higher in the HFD group than in the NFD group. In contrast, the mRNA expression levels of PPARγ, SREPB1c, FAS, and DGAT1 are markedly downregulated in the 10 × 109 CUF/day of L. plantarum ATG-K2 and Orlistat groups (Fig.3A). The mRNA expression levels of fatty acid oxidation-related genes, including PPARα and CPT1, and WAT browning-related genes, UCP1, PGC1α, Dio2, and ND5, are downregulated in the HFD group compared with the NFD group. These genes, PPARα and CPT1, UCP1, Prdm16, and ND5, are upregulated following treatment with 10 × 109 CFU/day of L. plantarum ATG-K2 and Orlistat groups compared with the HFD group (Fig. 3B and C).
L193-195 eliminate “tend to be higher”, this means that the expression level was the same or not affected.
- Response
Thank you for your comments. As your comments, it was corrected as follow (line 214-216).
2.8. Effects of L. plantarum ATG-K2 on inflammation markers in intestine and serum
As shown in Fig.4A, there was no difference on mRNA expression levels of TNF-α and IL-6 in the intestine between NFD and HFD groups.
L212-213 eliminate the results related with a “tendency”, this is not an effect
L213-215 The same comments for this sentence.
- Response
- Thank you for your comments. As your comments, it was corrected in revised manuscript as following (line 226-246).
2.9. Changes of gut microbiota by L. plantarum ATG-K2
To examine the effects of L. plantarum ATG-K2 on the gut microbiota, the bacterial community from cecal samples of each experimental group was analyzed. The results of the family level relative abundance analysis are shown in Fig.5A. The family Lactobacillaceae showed a significant increase in the L. plantarum ATG-K2 and NFD groups. The family Akkermansiaceae exhibited a significant increase in the Orlistat group, and the L. plantarum ATG-K2 and NFD groups showed no change. The families Bacteroidaceae, and Helicobacteraceae showed a significant increase in the HFD group compared with the NFD group but was not changed in L. plantarum ATG-K2 and Orlistat groups. The family Rikenellaceae showed a significant decrease in the Orlistat groups. The family Peptococcaceae showed a significant decrease in the NFD and the Orlistat groups compared with the HFD group but was not in L. plantarum ATG-K2 group (Fig.5B). Furthermore, gut bacteria richness was evaluated using chao1, and the diversity was evaluated using the Shannon index and Simpson’s index (Supplementary Fig. 1). The richness and diversity indices are decreased in the HFD, L. plantarum ATG-K2, and the Orlistat groups compared with the NFD group. Beta-diversity was examined by principal coordinate analysis (PCoA) to compare microbiota composition using UniFrac distance. Weighted-UniFrac takes into account the relative abundance of species/taxa shared between samples. The microbiota from the NFD group was separated from those of the other groups. The L. plantarum ATG-K2 group exhibits relatively similar changes to the Orlistat group than the HFD group (Fig. 5C).
L217-219 the same comments for these results. Please check the Supp Fig 1, the panel do not correspond with the legend.
- Response
Thank you for your comment. It was corrected in revised Supp Fig 1.
Figure S1. The richness and diversity of L. plantarum ATG-K2 in HFD-induced obese mice. (A) Chao 1. (B) Shannon index. NFD: Normal fat diet; HFD: High-fat diet; 4x109, 10x1010: HFD+ 4x109 or 10x1010 CFU/day L. plntarum ATG-K2; Orlistat: HFD + Orlistat 15.6 mg/kg. The ends of the whiskers represent the minimum and maximum, the bottom and top of the box are the 1st and 3rd quartiles, and the line within the box is the median. Statistical significance was as follows: *p<0.05 and **p<0.01 vs HFD (n=8 per group).
L222-225 An statistical analysis must be performed to evaluate effect of the group in microbiota.
- Response
- Thank you for your comment. PCoA result was performed the statistical analysis.
L235-236 butyric acid is not lower in HFD compared to NFD
L238-239 eliminate the results related with a “tendency”, this is not an effect
- Response
Thank you for your comments. We re-performed the statistical analysis, and confirmed that butyric acid levels are lower in HFD compared with NFD. So, it was replaced with new version Fig.6B. (line 266). In addition, it was collected as follow
2.10. Effects of L. plantarum ATG-K2 on SCFAs levels in feces (line 259-261)
A significant increase in SCFAs, acetic acid, propionic acid, and valeric acid levels were observed in the feces of both L. plantarum ATG-K2 and Orlistat groups. Butyric acid levels were increased in the Orlistat groups.
How the beneficial effect of the gut microbiome on obesity may be attributable to the derived metabolites SCFAs and BSCFAs?
- Response
Thank you for your comment.
Gut microbiome hydrolyzes nondigestible carbohydrates into oligosaccharides and then monosaccharides, which they ferment in the anaerobic environment. Major bacterial metabolic routes for SCFA production are the Embden-Meyerh of Parnas pathway (glycolysis, for six-carbon sugars) and the pentose-phosphate pathway (for five carbon sugars), which convert monosaccharides into phosphoenolpyruvate (PEP). Subsequently, PEP is converted into fermentation products such as organic acids such as pyruvate. Pyruvate is converted to acetyl-CoA with the concomitant formation of H2 and CO2, and then SCFAs is produced via different pathway. Also, side-product such as H2 are utilized by other species in cross-feeding to avoid accumulation of H2 which would inhibit the ability of primary fermenters to oxidize NADH [Besten et al 2013]. Therefore, it suggests that gut microbiome contribute enzymes, and beneficial effect of the gut microbiome is favorable to SCFAs production.
Reference
- den Besten, G.; van Eunen, K.; Groen, A.K.; Venema, K.; Reijngound, D.J.; Bakker, B.M, The role of short-chain fatty acids in the interplay between diet, gut microbiota, and host energy metabolism. J Lipid Res. 2013, 54, 2325-2340.
Can we consider a dysbiosis the modification of the relative abundances of 4 bacterial families?
- Response
Thank you for your comment. Dysbiosis of gut microbiome include the disruption of normal gut microbiome, but also shift such as change abundance, diversity, and relative distribution of the gut microbiome composition. It is considered important that gut microbiome shift different from normal state with changed bacterial number. Furthermore, dysbiosis of gut microbiome is observed in obese state: decreases of Bifidobacterium spp. and Akkermansia, and increase of populations of Ruminococcaceae and Rikenellaceae during high-fat diet-induced obese [Cani, 2013; Okeke, 2014]. These dysbiosis of gut miccrobiome by HFD were observed in the present study, while L. plantarum ATG-K2 restored the these dysbiosis. Therefore, these results suggest that L. plantarum ATG-K2 regulates the gut microbiome, although it remains to be investigated in clinical trials.
The authors mentioned that the probiotic can restore the gut microbiota by increasing the abundance of benefic bacteria. Which ones? The Lactobacillaceae family? Can this increase be attributed to the administration of the Lactobacillus strain? Did you check similarities between the probiotic and Lactobacillae families? Which genera increased their abundances?
- Response
Thank you for your comments. It has been reported that Lactobacillacease family can produce a lactic acid, and lactic acid is used to produce the SCFAs, acetic acid and propionic acid. This suggests that change of abundance of Lactobacillaceae family has partially beneficial effect on gut microbiome composition. On the other hand, the increase of genus Lactobacillus and pediococcus among Lactobacillace family was observed, but abundance of genus pediococcus is near to 0. Additionally, the abundance of Lactobacillace family was increased in dose dependent manner. It is suggested that administration of L. plantarum ATG-K2 may be attributed to increase of Lactobacillacease family. However, it needs to investigate whether L. plantarum ATG-K2 attribute to abundances of Lactobacillacease family increase with specific primer to detect the L. plantarum ATG-K2.
Results indicate that microbiota identification was performed on cecal samples, but Methods indicate that the analysis was done in fecal samples. Please revise.
- Response
Thank you for your comment. As your comment, it was corrected in revised manuscript (line 258).
BSCFAs are the results of fermentation of the protein. Please discuss this point.
- Response
Thank you for your comment. It was discussed as follow.
It is known that BSCFA are mainly produced during fermentation of branched chain amino acids (valine, leucine, and isoleucine) by the gut microbiome. BSCFA production share the same multienzyme pathway with SCFA production, however, there is difference on substrates initiating the pathway. For SCFA production, acetyl-CoA serves as the substrate, whereas for BSCFA production, branched-chain acyl-CoA serves as the substrate. Branched-chain acyl-CoA is converted by branched-chain –α- keto acid dehydrogenase complex (BCKDH), and gut microbiome is involved in this process, although it is remained the study for identification of particular bacterial taxa responsible for BSCFA modulating effects [Gojda and Cahova, 2021]. In the present study, BSCFAs levels were increased by L. plantarum ATG-K2.
Reference
- Gojda, J.; Cahova, M, Gut microbiota as the link between elevated BCAA serum levels and insulin resistance. Biomolecules 2021, 11, 1414.
- Responses to comments of “How the beneficial effect of the gut microbiome on obesity may be attributable to the derived metabolites SCFAs and BSCFAs?” and “BSCFAs are the results of fermentation of the protein” was discussed as follow (line 329-336).
SCFAs are generated through the colonic fermentation of dietary fibers and BSCFAs are generated by fermentation of branched amino acids by the gut microbiome, respectively. Their production shared the same multienzyme pathway; however, there is a difference in substrate initiation of the pathway. In addition, side-product such as H2 is utilized by other species in cross-feeding to avoid the accumulation of H2, which would inhibit the ability of primary fermenters [Besten et al 2013]. It suggests that the gut microbiome contributes enzymes with cross-feeding, and the beneficial effect of the gut microbiome is favorable to SCFAs and BSCFAs production. However, it remains the study for identification of particular bacteria responsible for their modulating effects [Gojda and Cahova, 2021].
L309-314 please include the concentration of BSCFAs that induce these effects, and compare to those concentrations determined in fecal samples.
- Response
Thank you for your comment. Based on report that effect of BSCFAs on glucose and lipid metabolism in adipocytes, these effects was observed at 10 mM BSCFAs. On the other hand, we analyzed the BSCFAs in feces of mice, reflects the concentrations of these compounds at the digestive tract. But, one of the limitations is difficult to compared to concentration between in vitro and in vivo, since absorption processes are also implied.
L282-289 The inhibition of fat absorption can be exerted by inhibiting the pancreatic lipase, such as orlistat, or at the intestinal level. Please specify which mechanisms are you discussing here.
- Response
Thank you for your comment.
We measured only fecal TG levels, since we observed the lipid layer in cecum in the process of DNA extract for analysis of cecal bacteria community as like orlistat. In addition, it has been reported that L. gasseri lowers transport rate of fatty acids in intestine [Hanmmed et al., 2009; Sato et al., 2008]. In our study, L. plantaun ATG-K2 tended to increase the fecal TG levels, which suggest a possibility to inhibit the fat absorption in intestine. As your comment, it needs to investigate the underlying mechanism involved in inhibition of fat absorption.
References
- Hamad, E. M.; Sato, M.; Uzu, K.; Yoshida, T.; Higashi, S.; Kawakami, H.; Kadooka, Y.; Matsuyama, H.; Abd El-Gawad, I. A.; Imaizumi, K., Milk fermented by Lactobacillus gasseri SBT2055 influences adipocyte size via inhibition of dietary fat absorption in Zucker rats. Br J Nutr 2009, 101, (5), 716-724.
- Sato, M.; Uzu, K.; Yoshida, T.; Hamad, E. M.; Kawakami, H.; Matsuyama, H.; Abd El-Gawad, I. A.; Imaizumi, K., Effects of milk fermented by Lactobacillus gasseri SBT2055 on adipocyte size in rats. Br J Nutr 2008, 99, (5), 1013-7.
L351-352 How the probiotic pellet was treated with lysozyme? Did you resuspend the pellet in PBS? What concentration of lysozyme did you used?
- Response
Thank you for your comment. The method for lysate of L.plantarum ATG-K2 is as follow and was replaced in Materials and methods section (line 379-391)
4.1 Preparation of L. plantarum ATG-K2
Briefly, L. plantarum ATG-K2 isolated from Korean fermented cabbage (Korean Collection for Type Culture [KCTC 13577BP]) was incubated in De Man Rogosa Sharp broth (Difco Laboratories Inc., NJ, USA) at 37 ℃ for 16 h, then cell pellets were obtained by centrifugation (3,000 × g, 10 min, 4 ℃), and washed three times with phosphate-buffered saline (PBS; pH7.4). To prepare the L. plantarum ATG-K2 lysates for in vitro experiments, the cell pellets were concentrated to 10X (1 x 1010 CFU/ml) by resuspending in PBS and were treated with 10 mg/ml lysozyme (Sigma-Aldrich, St. Louis, MO, USA) at 37 ℃ for 2h, and then were lysed with sonication. Stock solution (50 mg/ml) was prepared after solid content of resulting lysate was determined by a moisture analyzer (A&D Co., Ltd, Tokyo, Japan). For in vivo experiments, the cell pellets were resuspended in cryoprotectant solution and lyophilized using an FD8508 freeze-dryer (ilShinBioBase, Dongduchen, Korea). The freeze-dried L. plantarum ATG-K2 powder was resuspended in PBS and prepared daily for animal experiment periods.
What was the purpose to prepare a probiotic lysate if this was not used in experiments?
- Response
Recent studies have suggested that even dead probiotic cells or their cell components containing metabolites can be beneficial for host health. To determine the effect, probiotics lysate was used in vitro experiments including cell viability, adipogenesis, and genes and protein expression related adipogenesis in 3T3-L1 adipocytes. To make this clear, it was added in the Result section. Please refer section 2.1.~ 2.3.
Minor comment:
Line 49 “spp.” should not be italized
- Response
- Thank you for your comment. As you comment, it was corrected (line 50).
L362 what is “anti-anti”
- Response
- “anti-anti” means “Antibiotic-Antimycotic”, the commercial name of cell culture reagent. Please refer line 393.
Please use the name of the chemical compound instead of the commercial brand Xenical.
- Response
- As you comment, commercial brand Xenical was replaced with Orlistat in manuscript and figure, and figure legends.
Reviewer 3 Report
[General comments]
Obesity is the cause of life-related diseases and uncontrolled consumption of fat-containing foods is accelerating the increase of obesity. Pharmacological and surgical treatments are conducted for the obesity patients, but there are some risks in these methods. Therefore, it is important to develop the safe and effective ways to prevent obesity. As it is generally known that obesity is closely connected with the dysbiosis of gut microbiota, improvement of gut microbiota by probiotics may be effective. Although probiotics are live microorganisms, their safety is warranted by eating experience. L. plantarum ATG-K2 has been isolated from the Korean traditional fermented food and therefore it is reasonable to use this bacterium for the treatment of obesity. The concept is agreeable. However, there are several issues to be overcome for the practical application.
[Specific comments]
- Line 17: the effect of probiotics on the anti-obesity effect remains unknown. ⇒ the anti-obesity effect of probiotics remains unknown. (“effect” is redundant.)
- Line 18-19: Lactobacillus plantarum ⇒ Lactiplantibacillus plantarum (Refer to Zheng J, et al. Int J Syst Evol Microbiol. 70: 2782–2858, 2020).
- Line 73-75: To evaluate the cytotoxic effect of plantarum ATG-K2, 3T3-L1 cells were cultured for 72 h (3 days). On the contrary, to assess its anti-lipid accumulating effect, 3T3-L1 cells were cultured for 7 days. Why did not the authors check the viability of 3T3-L1 cells after 7 days’ culture?
- Line 118-128: The data showed that 10 x 109, but not 4 x 109, of plantarum ATG-K2 were effective for decreasing the body weight of HFD-fed mice. Do the authors calculate which dose of L. plantarum ATG-K2 are necessary for reducing the body weight in humans?
- Table 2: It is important that the authors compared the effects of plantarum ATG-K2 with Xenical, considering the application for the treatment of obesity. Although I imagine that Xenical (lipase inhibitor) may reduce the lipid-related substance in serum, L. plantarum ATG-K2 reduced serum TG but Xenical did not. How could we understand this result?
- Line 191: In the section title, “histological morphology of intestine” is included. However, there appears to be no data and description about the morphology of intestine in this section. What happens in the intestine?
- Figure 5: The effect on gut microbiota is interesting. It is reasonable to observe the improvement of gut microbiota by the administration of plantarum ATG-K2. Xenical also affected the gut microbiota, and PCA plot shows that the effects of L. plantarum ATG-K2 and Xenical are overlapping. These results suggest that the change of gut microbiota is not the cause but the effect of improved fat metabolism. How do the authors interpret this result?
- Line 349-350: How many percentage of freeze-dried plantarum ATG-K2 are alive?
Author Response
Comments and Suggestions for Authors
[General comments]
Obesity is the cause of life-related diseases and uncontrolled consumption of fat-containing foods is accelerating the increase of obesity. Pharmacological and surgical treatments are conducted for the obesity patients, but there are some risks in these methods. Therefore, it is important to develop the safe and effective ways to prevent obesity. As it is generally known that obesity is closely connected with the dysbiosis of gut microbiota, improvement of gut microbiota by probiotics may be effective. Although probiotics are live microorganisms, their safety is warranted by eating experience. L. plantarum ATG-K2 has been isolated from the Korean traditional fermented food and therefore it is reasonable to use this bacterium for the treatment of obesity. The concept is agreeable. However, there are several issues to be overcome for the practical application.
[Specific comments]
Line 17: the effect of probiotics on the anti-obesity effect remains unknown. ⇒ the anti-obesity effect of probiotics remains unknown. (“effect” is redundant.)
- Response
- Thank you for your comment. As you comment, it was corrected (line 18).
Line 18-19: Lactobacillus plantarum ⇒ Lactiplantibacillus plantarum (Refer to Zheng J, et al. Int J Syst Evol Microbiol. 70: 2782–2858, 2020).
- Response
- Thank you for your comment. As you comment, Lactobacillus plantarum was corrected to Lactiplantibacillus plantarum in manuscript including title (line19, 34, and 76).
Line 73-75: To evaluate the cytotoxic effect of plantarum ATG-K2, 3T3-L1 cells were cultured for 72 h (3 days). On the contrary, to assess its anti-lipid accumulating effect, 3T3-L1 cells were cultured for 7 days. Why did not the authors check the viability of 3T3-L1 cells after 7 days’ culture?
- Response
- Thank you for your comment. Since the doubling time of 3T3-L1 cells was approximately 18 h, we experimented with cell viability assays for 72 h, which is sufficient time to assess non-toxic concentrations. As shown in Fig. 1C, cytotoxic effect by L. plantarum ATG-K2 was not observed during the differentiation period, which confirmed with RNA and Protein quantitative values.
Line 118-128: The data showed that 10 x 109, but not 4 x 109, of plantarum ATG-K2 were effective for decreasing the body weight of HFD-fed mice. Do the authors calculate which dose of L. plantarum ATG-K2 are necessary for reducing the body weight in humans?
- Response
Thank you for tour comment. FDA guidelines “Guidance for Industry: Estimating the maximum safe starting dose in initial clinical trials for therapeutics in adult healthy volunteers (2005)” cannot be used to choose the dose of probiotics in human. Recently, it has been suggested calculation methods considering the intestinal surface area between animals and human. But it needs to study. In addition, according to many reports, most of efficacy experiments of probiotics in vivo were performed at dose of 1x108~1x1011. We performed the preliminary study for dose range finding and we selected the dose (4x109 and 10x109 CFU/day/mouse).
Table 2: It is important that the authors compared the effects of plantarum ATG-K2 with Xenical, considering the application for the treatment of obesity. Although I imagine that Xenical (lipase inhibitor) may reduce the lipid-related substance in serum, L. plantarum ATG-K2 reduced serum TG but Xenical did not. How could we understand this result?
- Response
Thank you for tour comment. As your comment, Orlistat (Xenical) may reduce the serum TG, but this effect was not observed in the present study. This result is consistent with previous effect [Choi et al., 2017; Lee et al., 2019]. This could be because that concentration of orlistat used in the study is 15.6 mg/kg, which is a low concentration as compared to the commonly used concentration of 10–50 mg/kg.
Reference
- Choi, J.; Kim, K.J.; Koh, E.J.; Lee, B.Y., Gelidium elegans regulates the AMPK-PRDM16-UCP-1 pathway and has a synergistic effect with orlistat on obestiy-associated features in mice fed a high-fat diet. Nutrient 2017, 9, 342.
- Lee, M.R.; Kim, J.E.; Choi, J.Y.; Park, J. J.; Kim, H. R.; Song, B.R.; Choi, Y.W.; Kim, K.M.; Song, H.; Hwang, D.Y., Anti-obesity effect in high-fat-diet-induced obese C57BL/6 mice: Study of a novel extract from mulberry (Morus alba) leaves fermented with Cordyceps militaris. Exp Ther Med. 2019, 17, 2185-2193.
Line 191: In the section title, “histological morphology of intestine” is included. However, there appears to be no data and description about the morphology of intestine in this section. What happens in the intestine?
- Response
Thank you for your comment. It is typing mistaking. It was deleted in revised manuscript.
Figure 5: The effect on gut microbiota is interesting. It is reasonable to observe the improvement of gut microbiota by the administration of plantarum ATG-K2. Xenical also affected the gut microbiota, and PCA plot shows that the effects of L. plantarum ATG-K2 and Xenical are overlapping. These results suggest that the change of gut microbiota is not the cause but the effect of improved fat metabolism. How do the authors interpret this result?
- Response
Thank you for your comment.
Gut microbiome ferments the carbohydrates and produces the its metabolites such as short chain fatty acids (SCFAs), and maintains intestinal permeability, which exert metabolically beneficial effects on host. Therefore, interaction between diet and gut microbiome in intestine is an important modulator of host metabolism (homeostasis). However, high fat-diet induces gut microbial alternation, which plays a key role in driving these metabolic disorders through the following potential mechanisms: increased intestinal permeability, increased metabolic endotoxemia such as LPS, increased systemic inflammation, and reduced SCFAs. Orlistat (Xenical) is a gastric and pancreatic lipase inhibitor and inhibit the absorption of ingested fat. Ke et al., reported that Orlistat changes the microbial composition and promotes functional shifts in the gut microbiota in HFD-obese mice. This report suggests that anti-obesity effect of orlistat may exert by modifying gut microbiome, and offer a novel mechanism of it. Therefore, these results suggest that the anti-obesity effect of L. plantarum ATG-K2 may be influenced by modulation of gut microbiome.
Reference
- Ke, J.; An,Y.; Cao,B.; Lang, J.; Wu, N.; Zhao, D., Orlistat-induced gut microbiota modification in obese mice. Evid Based Complement Alternat Med. 2020, 2020, 9818349.
Line 349-350: How many percentage of freeze-dried plantarum ATG-K2 are alive?
- Response
The survival rate of L. plantarum ATG-K2 after lyophilization is a about 70%. Based on the survival rate after freeze-drying, the administration dose in vivo was calculated.
Round 2
Reviewer 2 Report
The authors have corrected several sections of the manuscript. However, there are still several points that must be revised. The rationale to use the concentrations of the lysate probiotic in the in vitro studies is missing.
- In the introduction section, please explain the rationale to select a pharmacological approach to treat obesity instead of a dietary intervention.
L60-62 please revise. This sentence has several errors. Please revise the literature and be precise. The effect is induced by the HFD. “dysbiosis induces LPS production from Gram-negative bacteria” LPS are normal structural molecules of Gram- Negative bacteria. Please check the exact effect of HFD on Gram-negative bacteria.
L64-65 - “we also observed” these are previous work of the authors?
- Please revise the last revisions related with validity to use the ratio of Firmicutes/Bacteroidetes as a hallmark of obesity. This ration has been extremely questioned.
- The descriptions of the modification of the microbiota in obesity and HFD are based on 2 reviews (11, 12) of the first studies. Please update this information. Please revise updated metaanalysis on the field.
L71-73 this scientific evidence is based on a review (14). Please use the appropriate reference. Also, please revise the reference 14, which is very precise to point that there is still a debate to identify signature in healthy or obese microbiota composition, and the role of an altered gut microbiome as a cause or consequence of obesity in human studies is very controversial. According to that, I expect that authors exposed the complete overview of the field, and not just convenient evidence.
L75-77 what was the effect of these 2 probiotics on obesity?
L78-80 - “…is an attractive strategy” to what?
- “The survival rate of plantarumATG-K2 after lyophilization is a about 70%. Based on the survival rate after freeze-drying, the administration dose in vivo was calculated”. Please include this information in methods section 4.6
- The same question again. It is necessary that authors justify in the manuscript what they select these concentration of lysate probiotic to be test in in vitro experiments. Can these lysate concentrations reach these cells in in vivo conditions? The same comment for lipid accumulation, expression, AMPK activation analysis.
L146-149 Authors included the include energy consumed by each group of animals that can explain the higher weight gain in HFD group, despite a lower food intake. However, the interpretation of this result is incorrect. How a lower intake of energy induce a higher weight gain? Please carefully revise the NFD and HFD energy data in Table 1.
- Again, please check the Supp Fig 1, the panel do not correspond with the legend.
L260-262 The richness and diversity indices were not affected.
L265-268 Again, what was the result of the statistical analysis on PCoA? Do you detect statistical differences in the microbiota of the different groups? It is not enough to describe if the groups are separated or not in the PCoA.
L375-383 please check the English redaction of these sentences. The meaning is not clear.
- The authors mentioned that the probiotic can restore the gut microbiota by increasing the abundance of benefic bacteria. Which ones? The Lactobacillaceae family? Can this increase be attributed to the administration of the Lactobacillus strain? Did you check similarities between the probiotic and Lactobacillae families? Which genera increased their abundances? Please discuss these points in the discussion section.
- Results indicate that microbiota identification was performed on cecal samples, but Methods indicate that the analysis was done in fecal samples. Please revise.
Author Response
Reviwer#2 Comments and Suggestions for Authors
The authors have corrected several sections of the manuscript. However, there are still several points that must be revised. The rationale to use the concentrations of the lysate probiotic in the in vitro studies is missing.
- In the introduction section, please explain the rationale to select a pharmacological approach to treat obesity instead of a dietary intervention.
- Response
Thank you for your comment. It was added into revised manuscript as follow (line 41~44).
Obesity is an extensive public health problem, and its prevalence has been increasing over the last few decades [1]. It has been known as a risk factor of chronic metabolic syndromes, such as hyperlipidemia, type 2 diabetes, and cardiovascular disease; therefore, obesity management is important to prevent obesity related complications [1]. A dietary intervention including energy restriction, macronutrients, food, and dietary intake patterns, is recommended for treatment the obesity. Despite their effectiveness, a dietary intervention remains an unachieved weight loss [Scheen and Lefébvre, 1999; Fujioka et al., 2002; Higuera-Hernández et al., 2018]. Currently, various pharmacological approaches are used to treat obesity, and their use is limited because of their undesirable side effects.
Reference
- Scheen, A.J.; Lefébvre, P.J., Pharmacological treatment of obesity: present status, Int J Obes Relat Metab Disord 1999, 23, 47-53.
- Fujioka, K., Management of obesity as a chronic Disease: Nonpharmacologic, pharmacologic, and surgical options, Obes Res 2002, 10,116s-123s.
- Higuera-Hernández, M.F., Reyes-Cuapio, E., Gutiérrez-Mendoza, M., Rocha, N.B., Veras, A.B., Budde, H., Jesse, J., Zaldívar-Rae, J., Blanco-Centurión, C., Machado, S., Murillo-Rodríguez. E., Fighting obesity: Non-pharmacological interventions. Clin Nutr ESPEN 2018, 25, 50-55.
L60-62 please revise. This sentence has several errors. Please revise the literature and be precise. The effect is induced by the HFD. “dysbiosis induces LPS production from Gram-negative bacteria” LPS are normal structural molecules of Gram- Negative bacteria. Please check the exact effect of HFD on Gram-negative bacteria.
- ResponseIn addition, it was corrected as follow (line 73~75).
- L64-65 - “we also observed” these are previous work of the authors?
- Dysbiosis increase intestinal permeability and translocation of LPS into target tissues, which causes metabolic endotoxemia. Consequently, metabolic endotoxemia induces low-grade chronic inflammation in obesity and its complications through a Toll-like receptor-mediated inflammatory pathway [18].
- Thank you for your comment. As your comments, the sentence “dysbiosis induces LPS production from Gram-negative bacteria” was deleted in revised manuscript. According to reports, HFD reduces the Gram-negative bacteria Bacteroides and Gram-positive bacteria Bifidobaterium spp.. Bifidobacteria can reduce the plasma levels of endotoxins by improving gut barrier. In addition, LPS produced in the gut by the death of Gram-negative bacteria transported form the intestinal toward target tissues by lipoproteins, chylomicrons synthesized form epithelial intestinal cells in response to the HFD. Based on these reports, it is considered that dysbiosis and/or HFD induced metabolic endotoxemia vis regulation of gut permeability and translocation into target tissue, but not production of LPS.
- Response
- There was typing mistake. However, it was deleted in revised manuscript, since introduction was replaced as your comment.
- Please revise the last revisions related with validity to use the ratio of Firmicutes/Bacteroidetes as a hallmark of obesity. This ration has been extremely questioned.
- The descriptions of the modification of the microbiota in obesity and HFD are based on 2 reviews (11, 12) of the first studies. Please update this information. Please revise updated metaanalysis on the field.
L71-73 this scientific evidence is based on a review (14). Please use the appropriate reference. Also, please revise the reference 14, which is very precise to point that there is still a debate to identify signature in healthy or obese microbiota composition, and the role of an altered gut microbiome as a cause or consequence of obesity in human studies is very controversial. According to that, I expect that authors exposed the complete overview of the field, and not just convenient evidence.
- Response
Thank you for your comments. As your comments, it was replaced as follow (line 56~79).
Dysbiosis or imbalance of the gut microbiome is associated with obesity and its complications [9]. Dysbiosis or imbalanced gut microbiome means disruption of balance between the gut microbiota and host and reflects negative shifts in abundance, diversity, and relative distribution of the gut microbiome composition [6, 10]. Both in animal and human studies with obesity, altered gut microbiome with a reduction in gut microbiome diversity and richness has been reported. Consistent finding has shown that obesity is associated with a decrease abundance in some taxa such as Bifidobaterium, Christensenellacease, and Akkermansia, however, there are that the change of gut microbiome reported to differ with obesity have varied across studies. For example, the Firmicutes/Bacteroidetes ratio in obese human compared with the lean human has been reported to decrease, to increase, or not change at all [12, Hermes et al., 2015; Singer-Englar et al., 2018]. These inconsistencies may be affected by a combination of large interpersonal variation, insufficient sample sizes, and methodological differences between studies. There is also observed that change of functional include a regulation in production of enzymes involved in carbohydrate and lipid metabolism in obese mice. Overall, most studies have demonstrated a structural and functional dysbiosis of gut microbiome in obesity, although there is still much debate on the role of gut microbiome in obese human [Jiao et al., 2018; 15]. On the other hands, the gut microbiome plays an important role in the carbohydrate fermentation, and they produce short-chain fatty acids (SCFAs), such as acetic, propionic, and butyric acids, which critical roles in preventing and treating obesity and its complications by improving glucose and lipid homeostasis and decreasing inflammation [14].
References
1.Singer-Englar, T., Barlow, G., Mathur, R., Obesity, diabetes, and the gut microbiome: an update review. Expert Rev Gastroenterol Hepatol 2018, 13, 3-15.
2.Hermes, G.D.A., Zoetendal, E.G., Smidt, H., Molecular ecological tools to decipher the role of our microbial mass in obesity., Benef Microbes 2015, 6, 61–81.
- 3. Jiao, N., Baker, S. S., Nugent, C. A., Tsompana, M., Cai, L.,Wang, Y., Buck, M. J., Genco, R. J., Baker, R. D., Zhu, R., Zhu, L., Gut microbiome may contribute to insulin resistance and systemic inflammation in obese rodents: a meta-analysis. Physiol Genomics2018, 50(4), 244-254.
L75-77 what was the effect of these 2 probiotics on obesity?
- Response For example, L. rhamnous LS-8 and L. crustorum MN407 showed the anti-obesity effect with reduced body weight gain, insulin resistance, and inflammatory, but also manipulated gut microbiota by decreasing the abundance of Bacteroides and Desulfovibrio and increasing Lactobacillus and Bifdobacterium, which led to increasing the levels of SCFAs in feces [7].
- Thank you for your comments. 2 probiotics is no observed in L75-77. But, there is observed in Line 71. It was replaced as follow (Line 81~86).
L78-80 - “…is an attractive strategy” to what?
- Response As described in manuscript, it is considered that targeting the regulation of gut microbiome is attractive strategy.
- Thank you for your comment. “attractive” was deleted in revised manuscript (line 85~86).
- “The survival rate of L. plantarumATG-K2 after lyophilization is a about 70%. Based on the survival rate after freeze-drying, the administration dose in vivo was calculated”. Please include this information in methods section 4.6.
- Response
- It was added into revised manuscript (line 398~407).
For in vivo experiments, the cell pellets were resuspended in cryoprotectant solution and lyophilized using an FD8508 freeze-dryer (ilShinBioBase, Dongduchen, Korea). Based on the survival rate after freeze-drying, the administration dose in vivo was calculated. To determine the survival rate, the freeze-dried L. plantarum ATG-K2 powder was resuspended in 0.9% sterilized saline and prepared daily for animal experiment periods. To analyze, the freeze-dried L. plantarum ATG-K2 powder was suspended in saline, and the was serially diluted to 10-8. After a serial dilution, diluted L. plantarum ATG-K2 powder was inoculated onto MRS agar and incubated in 37 ℃ for 24h. The colony number was counted, and colony forming unit (CFU) was calculated by colony number x dilution factor.
- The same question again. It is necessary that authors justify in the manuscript what they select these concentration of lysate probiotic to be test in in vitro experiments. Can these lysate concentrations reach these cells in in vivo conditions? The same comment for lipid accumulation, expression, AMPK activation analysis.
- Response
- In most of the functional studies using natural materials, screening and mechanism of their functions are assessed by in vitro assays. In general, the maximum concentration of natural material without toxicity is applied in cell experiments, and we also followed the same procedure. Basically, we have optimized and validated analytical conditions such as solubility and incubation time through a series of preliminary experiments. Since the concentration used in the cell experiment cannot be applied in vivo, it is necessary to set a new concentration in the animal experiment. The concentration of animal experiment using L. plantarum ATG-K2 were selected by referring Kwon et al and preliminary experiment. We think it played a role since lipid accumulation, expression, AMPK activation analysis of 3T3-L1 were confirmed with in vivo phenotype results and related gene expression.
References
Kwon, J.; Kim, B.; Lee, C.; Joung, H.; Kim, B.K.; Choi, I.S., Hyun, C.K., Comprehensive amelioration of high-fat diet-induced metabolic dysfunction through activation of the PGC-1a pathway by probiotics treatment in mice. PLoSONE 2020, 15, e0228932.
L146-149 Authors included the include energy consumed by each group of animals that can explain the higher weight gain in HFD group, despite a lower food intake. However, the interpretation of this result is incorrect. How a lower intake of energy induce a higher weight gain? Please carefully revise the NFD and HFD energy data in Table 1.
- Response Food intake in the HFD group was lower than that in the NFD and Orlistat groups. Energy intake in the NFD group was decreased compared with the HFD group, but was increased in the Orlistat group. However, the L. plantarum ATG-K2 group did not affect food intake and energy intake.
- Thank you for your comment. As your comment, it was corrected as follow (line 145~148).
Additionally, in the present study, food intake in the NFD group were higher, but energy intake and FER in the NFD group lower than those of the HFD group. Lower FER implies that less efficient in transforming the contained nutrients into own biomass, and higher FER means that weight gain in high even if eating less.
- Again, please check the Supp Fig 1, the panel do not correspond with the legend.
- Response
Thank you for your comment. It was corrected in revised Supp Fig 1.
Figure S1. The richness and diversity of L. plantarum ATG-K2 in HFD-induced obese mice. (A) Chao 1, (B) Shannon index, and (C) Simpson’s index. NFD: Normal fat diet; HFD: High-fat diet; 4x109, 10x109: HFD+ 4x109 or 10x109 CFU/day L. plantarum ATG-K2; Orlistat: HFD + Orlistat 15.6 mg/kg. The ends of the whiskers represent the minimum and maximum, the bottom and top of the box are the 1st and 3rd quartiles, and the line within the box is the median. Statistical significance was as follows: *p<0.05 and **p<0.01 vs HFD (n=8 per group).
L260-262 The richness and diversity indices were not affected.
- Response
- As no significant difference was observed in richness and diversity, we corrected the sentence (Line 243~244 ).
L265-268 Again, what was the result of the statistical analysis on PCoA? Do you detect statistical differences in the microbiota of the different groups? It is not enough to describe if the groups are separated or not in the PCoA.
- Response
- Thank you for your comment. We added the result of statistical analysis on PCoA as Supplementary Fig. 2. And added at Line 249~253.
L375-383 please check the English redaction of these sentences. The meaning is not clear.
- ResponseTo prepare the L. plantarum ATG-K2 lysates for in vitro experiments, the cell pellets were concentrated to 10 x by resuspending in PBS and were treated with 10 mg/ml lysozyme (Sigma-Aldrich, St. Louis, MO, USA) at 37 ℃ for 2h, and then were lysed with sonication. The resulting lysates (0.1g) was used to measure the dry mass. Dry mass was determined by using a moisture analyzer that combines drying and on-line weight. The remaining resulting lysate mass was adjusted to 50mg/mL to make a final stock solution concentration, and then it was used for in vitro experiment.
- Thank you for your comment. As your comment, it was collected in revised manuscript as follow (line 392~398).
- The authors mentioned that the probiotic can restore the gut microbiota by increasing the abundance of benefic bacteria. Which ones? The Lactobacillaceae family? Can this increase be attributed to the administration of the Lactobacillus strain? Did you check similarities between the probiotic and Lactobacillae families? Which genera increased their abundances? Please discuss these points in the discussion section.
- Response
- Thank you for your comment. As your comment, it was added into Discussion section in revised manuscript as follow (line 353~362).
It has been reported that Lactobacillacease family can produce a lactic acid, and lactic acid is used to produce the SCFAs, acetic acid and propionic acid. This suggests that change of abundance of Lactobacillaceae family has partially beneficial effect on gut microbiome composition. On the other hand, the increase of genus Lactobacillus and pediococcus among Lactobacillace family was observed, but abundance of genus pediococcus is near to 0, while the abundance of Lactobacillace family was increased in dose dependent manner. It is suggested that administration of L. plantarum ATG-K2 may be attributed to increase of Lactobacillacease family. However, it needs to investigate whether L. plantarum ATG-K2 attribute to abundances of Lactobacillacease family increase with specific primer to detect the L. plantarum ATG-K2.
- Results indicate that microbiota identification was performed on cecal samples, but Methods indicate that the analysis was done in fecal samples. Please revise.
- Response
- Thank you for your comment. As your comments, it was corrected. (Line 501)
Reviewer 3 Report
The manuscript has been well revised and the responses to my inquiries were almost agreeable. Why I asked the effective dose of L. plantarum ATG-K2 for humans, 10 x 10exp9 for mouse was considered to be higher compared with other known probiotics. If we calculate simply with body weight base, 10 x 10exp12 dose is necessary for humans. This amount is not practical. Of course, this speculation is over-simplified and may not be correct. So, I recommend that the authors will decide how many L. plantarum ATG-Ks is sufficient for humans' health.
Author Response
Reviwer#3 Comments and Suggestions for Authors
The manuscript has been well revised and the responses to my inquiries were almost agreeable. Why I asked the effective dose of L. plantarum ATG-K2 for humans, 10 x 10exp9 for mouse was considered to be higher compared with other known probiotics. If we calculate simply with body weight base, 10 x 10exp12 dose is necessary for humans. This amount is not practical. Of course, this speculation is over-simplified and may not be correct. So, I recommend that the authors will decide how many L. plantarum ATG-K2 is sufficient for humans' health.
- Response
- Thank you for your comment. As your comment, it needs to study about dose of L. plantarum ATG-K2 for human’ health. We suggest that dose of L. plantarum ATG-K2 in first-in-human is used as same dose of animal based on reporters and referring Table. Reports was used the same dose in vivo and in human.
Efficacy |
Probiotics |
In vivo dose |
In human dose |
Reference |
Anti-obesity |
Lactobacillus gasseri BNR17 |
1x109 or 1x1010CFU/day |
1x109 or 1x1010CFU/day |
Kang JH et al., 2013 Yun SI et al., 2009 Kim J et al., 2018 |
Lactobacillus curvatus HY7601 + Lactobacillus plantarum KY1032 |
1x1010CFU/day |
1x1010CFU/day |
Jeung WH et al., 2019 Park DY et al, 2013 Jung S et al., 2015 |
Reference
- Kang, J.H., Yun, S.I., Park, M.H., Park, J.H., Jeong, S.Y., Park, H.O., Anti-obesity effect of Lactobacillus gasseri BNR17 in high-sucrose diet-induced obese mice, PLoS One, 2013, 8, e54617.
- Yun, S.I., Park, H, O, Kang, J.H., Effect of Lactobacillus gasseri BNR17 on blood glucose levels and body weight in a mouse model of type 2 diabetes, J Appl Microbiol, 2009, 107(5):1681-6.
- Kim, J.H., Yun, J.M., Kim, M. K., Kwon, O., Cho, B., Lactobacillus gasseri BNR17 Supplementation Reduces the Visceral Fat Accumulation and Waist Circumference in Obese Adults: A Randomized, Double-Blind, Placebo-Controlled Trial, J Med Food, 2018 May;21(5):454-461.
- Jeung, W.H., Nam, W., Kim, H.J., Kim, J.U., Nam, B., Jang, S.S., Lee, J.L., Sim, J.H., Park,S. D., Oral Administration of Lactobacillus curvatus HY7601 and Lactobacillus plantarum KY1032 with Cinnamomi Ramulus Extract Reduces Diet-Induced Obesity and Modulates Gut Microbiota, Prev Nutr Food Sci, 2019, 24(2):136-143.
- Park, D.Y., Ahn, Y.T., Park, S.H., Huh, C.S., Yoo, S.R., Yu, R., Sung, M.K., McGregor, R.A., Choi, M.S., Supplementation of Lactobacillus curvatus HY7601 and Lactobacillus plantarum KY1032 in Diet-Induced Obese Mice Is Associated with Gut Microbial Changes and Reduction in Obesity, PLos One, 2013, 8, e59470.
- Jung, S., Lee, Y.J., Kim, M., Kim, M., Kwak, J.H., Lee, J.W., Ahn, Y.T., Sim, J.H., Lee, J.H., Supplementation with two probiotic strains, Lactobacillus curvatus HY7601 and Lactobacillus plantarum KY1032, reduces fasting triglycerides and enhances apolipoprotein A-V levels in non-diabetic subjects with hypertriglyceridemia, Atherosclerosis, 2015, 241; 649-656.

Round 3
Reviewer 2 Report
The authors have included all the suggestions and the article can be accepted for publication.